# LSD1 defines the fiber type-selective responsiveness to environmental stress in skeletal muscle

**Hirotaka Araki[1,2], Shinjiro Hino[1]\*, Kotaro Anan[1], Kanji Kuribayashi[1], Kan Etoh[1], Daiki Seko[3,4], Ryuta Takase[1], Kensaku Kohrogi[1], Yuko Hino[1], Yusuke Ono[3], Eiichi Araki[2], Mitsuyoshi Nakao[1]\***

[1]Department of Medical Cell Biology, Institute of Molecular Embryology and Genetics, Kumamoto University, Kumamoto, Japan; [2]Department of Metabolic Medicine, Faculty of Life Sciences, Kumamoto University, Kumamoto, Japan; [3]Department of Muscle Development and Regeneration, Institute of Molecular Embryology and Genetics, Kumamoto University, Kumamoto, Japan; [4]Department of Molecular Bone Biology, Graduate School of Biomedical Sciences, Nagasaki University, Nagasaki, Japan

**\*For correspondence:**
s-hino@kumamoto-u.ac.jp (SH);
mnakao@gpo.kumamoto-u.ac.jp (MN)

**Competing interest:** The authors declare that no competing interests exist.

**Abstract** Skeletal muscle exhibits remarkable plasticity in response to environmental cues, with stress-dependent effects on the fast-twitch and slow-twitch fibers. Although stress-induced gene expression underlies environmental adaptation, it is unclear how transcriptional and epigenetic factors regulate fiber type-specific responses in the muscle. Here, we show that flavin-dependent lysine-specific demethylase-1 (LSD1) differentially controls responses to glucocorticoid and exercise in postnatal skeletal muscle. Using skeletal muscle-specific LSD1-knockout mice and in vitro approaches, we found that LSD1 loss exacerbated glucocorticoid-induced atrophy in the fast fiber-dominant muscles, with reduced nuclear retention of Foxk1, an anti-autophagic transcription factor. Furthermore, LSD1 depletion enhanced endurance exercise-induced hypertrophy in the slow fiber-dominant muscles, by induced expression of ERRγ, a transcription factor that promotes oxidative metabolism genes. Thus, LSD1 serves as an 'epigenetic barrier' that optimizes fiber type-specific responses and muscle mass under the stress conditions. Our results uncover that LSD1 modulators provide emerging therapeutic and preventive strategies against stress-induced myopathies such as sarcopenia, cachexia, and disuse atrophy.

## Editor's evaluation

This paper investigates the role of lysine-specific histone demethylase 1 (LSD1) in skeletal muscle responses to stresses. The findings link LSD1 to responses of skeletal muscle to glucorticoids and exercise in a fiber-type-dependent manner. The paper should be of broad interest to those studying muscle biology and physiology.

## Introduction

Skeletal muscle is the largest organ in humans, and the loss of muscle mass and physical activities are linked to the risks of metabolic disorders and aging-associated diseases. Skeletal muscle is composed of distinct types of myofibers. The slow-twitch fibers (type I) have high respiratory potential, and thus, a high endurance capacity, while the fast-twitch fibers (type II) have high glycolytic capacity and are specialized for dynamic movements (*Schiaffino and Reggiani, 2011*). In mice, type II fibers can be

subdivided into three groups, type IIA, type IIX, and type IIB, with the type IIA relatively more oxidative than the types IIX and IIB. These myofibers are subject to gain or loss, with different sensitivities in response to numerous environmental cues, such as nutritional conditions, hormones, drugs, and exercise (*Braun and Marks, 2015*; *Cartee et al., 2016*; *Hawley et al., 2014*). For example, the fast fibers are preferentially affected in glucocorticoid (GC)-induced muscle atrophy, which is associated with certain environmental and pathological conditions, such as starvation, GC drug therapy, sepsis, cachexia, acute diabetes, and hypoxia (*Braun and Marks, 2015*; *Schakman et al., 2013*; *Kuo et al., 2013*; *Przygodda et al., 2017*). Although GCs induce atrophy by enhancing protein catabolism via the autophagy and ubiquitin-proteasome system (*Sartori et al., 2021*), the molecular basis for the fiber type preference is still unclear. Another environmental condition that can remarkably alter muscle phenotypes is exercise. By increasing muscle mass, resistance, and endurance, exercise confers health benefits in the young, old, and sick individuals (*Cartee et al., 2016*). In particular, endurance exercise promotes the formation of oxidative fibers, thus increasing the oxidative capacity in the muscle, which can effectively improve systemic metabolism in diabetic patients and prevent disuse-induced atrophy (*Yan et al., 2011*; *Qaisar et al., 2016*).

Several chromatin factors promote phenotypic plasticity in the skeletal muscle. SMYD3, a histone methyltransferase, promotes GC-induced atrophy in mice by enhancing the expression of atrophy-associated genes, such as *myostatin* and *Met* (*Proserpio et al., 2013*). In addition, SETD7, another histone methyltransferase, and histone deacetylases-4 and -5 have essential roles in cachexia and denervation-induced atrophy, respectively (*Nayak et al., 2019*; *Luo et al., 2019*; *Moresi et al., 2010*). Moreover, DNA methyltransferase 3A is essential for muscle adaptation to endurance exercise by suppressing the oxidative stress response in the muscle (*Damal Villivalam et al., 2021*). These previous findings in literature have established that epigenetic factors are the integral components of muscle adaptation. However, factors that define the fiber type-specific responses to environmental factors have not been identified.

Lysine-specific demethylase-1 (LSD1 or KDM1A) is a flavin-dependent amine oxidase that in general removes mono- and di-methyl groups of histone H3 at lysine 4 (H3K4) and other protein substrates (*Hino et al., 2016*; *Shi et al., 2004*; *Nakao et al., 2019*), suggesting its contribution to specific biological processes. Despite ubiquitous expression in various tissue types, LSD1 plays essential roles in myogenic differentiation and muscle regeneration (*Choi et al., 2010*; *Scionti et al., 2017*; *Tosic et al., 2018*). Notably, we recently demonstrated that LSD1 promotes fast-like myogenic differentiation by epigenetically repressing slow fiber genes in C2C12 mouse myoblasts (*Anan et al., 2018*). Thus, the available evidence raises the possibility that LSD1 influences the adaptive phenotypes in the muscle under environmental cues. To verify this, we generated muscle-specific/inducible LSD1-knockout mice and examined how LSD1 loss affects muscle plasticity under environmental stresses. We found that LSD1 depletion exacerbated GC-induced fast fiber atrophy while augmenting endurance exercise-induced slow fiber hypertrophy. In these adaptive processes, LSD1 regulated distinct sets of target genes, together with unique transcription factors (TFs). Collectively, our current study reveals the mechanism through which LSD1 builds the epigenetic barriers that restrict adaptive responses in the muscles.

## Results

### Muscle-specific LSD1 depletion exacerbates dexamethasone-induced muscle atrophy

LSD1 is also known as KDM1A. For clarity, we use the word LSD1 for the protein and *Lsd1* for gene and mRNA. To investigate the role of LSD1 in muscle plasticity in vivo during postnatal life, we generated tamoxifen-inducible-, skeletal muscle-specific LSD1-knockout (LSD1-mKO) mice by crossing LSD1-floxed mice (LSD1^*flox/flox*) and transgenic mice harboring ACTA1-creERT2 (*McCarthy et al., 2012*; *Figure 1—figure supplement 1A*). This strategy facilitated LSD1 depletion in differentiating and differentiated muscle cells. We confirmed that LSD1 was depleted after tamoxifen administration in both fast-twitch fiber-dominant (tibialis anterior [TA], gastrocnemius [Gas], and extensor digitorum longus) and slow-twitch fiber-dominant (soleus [Sol]) tissues (*Figure 1—figure supplement 1B, C*). The magnitude of LSD1 depletion was comparable among these tissues. Marginal amounts of LSD1 were present after Cre induction, probably due to the expression in hematopoietic and vascular cells

that are resident in muscle tissues (*Kerenyi et al., 2013*; *Yuan et al., 2022*). In contrast, LSD1 expression in the heart, liver, and adipose tissues remained unaffected. Next, we depleted LSD1 in growing mice by administrating tamoxifen to pre-weaning mice (11 days of age) (*Figure 1—figure supplement 2A*). The expression of muscle fiber genes was largely unaffected, suggesting that LSD1 depletion in our system did not affect postnatal muscle growth (*Figure 1—figure supplement 2B*).

Dexamethasone (Dex), a synthetic GC, affects both the muscle mass and quality by inducing protein catabolism in fast fiber-dominant muscles (*Quattrocelli et al., 2019*; *Figure 1—figure supplement 3A–C*). To determine the involvement of LSD1 in muscle plasticity under environmental stress, we administered Dex in wild type (WT) and LSD1-mKO mice after knockout (KO) induction (*Figure 1A*). We found that the atrophy of TA and Gas muscles was enhanced in LSD1-mKO mice (*Figure 1B*), which was also evident from their reduced body weight (*Figure 1—figure supplement 4A*). Consistently, major hallmarks of muscle atrophy, including enhanced processing of autophagy protein LC3, increased expression of p62, attenuated phosphorylation of Akt, were pronounced in the LSD1-mKO muscle (*Figure 1—figure supplement 5A–C*). Another atrophy marker, 4E-BP phosphorylation, remained unaffected (*Figure 1—figure supplement 5D*). Of note, LSD1 depletion alone did not affect the muscle size and body weight (*Figure 1B*, and *Figure 1—figure supplement 3D, E*), indicating that the effects of Dex were enhanced by the absence of LSD1. In contrast, the Sol muscle, which was only modestly sensitive to Dex, and non-muscle tissues were unaffected by LSD1 depletion (*Figure 1B* and *Figure 1—figure supplement 4B*). Next, we evaluated the muscle performance of LSD1-mKO mice after Dex administration. LSD1 depletion led to weakened muscle strength as revealed by a whole limb grip test (*Figure 1C*), while LSD1 depletion alone was not sufficient to induce this phenotype (*Figure 1—figure supplement 3F*). In contrast, the running endurance was unaffected as assessed by a treadmill run (*Figure 1—figure supplement 4C*). These results suggest that LSD1 protects fast-dominant muscles from Dex-induced atrophy, thereby maintaining muscle strength.

## Oxidative fiber ratio is increased in the fast-dominant muscle in LSD1-mKO mice after Dex administration

Because the effects of LSD1 depletion were specific to fast-dominant muscles, we examined if the fiber-type composition was affected in the KO mice under Dex treatment. Immunostaining of the Gas muscle revealed that the area and number of oxidative (type I and IIA) fibers were higher in LSD1-mKO mice than in WT (*Figure 1D-F*, and *Figure 1—figure supplement 6A*). In contrast, the area and number of glycolytic (type IIB+IIX) fibers were decreased in LSD1-mKO (*Figure 1D–F*, and *Figure 1—figure supplement 6B*). Such genotype-dependent differences were not observed in the absence of Dex (*Figure 1—figure supplement 6C,D* and *Figure 1—figure supplement 7A,B*). In both genotypes, we did not observe centralized nuclei in the muscle fiber, which is characteristic of newly regenerating myofibers (*Figure 1—figure supplement 7C*). This suggests that the increase in the number of slow-twitch fibers in the KO was not due to enhanced de novo fiber formation. Additionally, we analyzed the size distribution of myofibers according to cross-sectional area and found that large type I and type IIA fibers preferentially increased in the LSD1-mKO muscle, whereas large type IIB+IIX fibers decreased (*Figure 1—figure supplements 6B and 7D and E*). Taken together, these results suggest that oxidative fibers are preferentially formed and/or maintained in LSD1-depleted muscles under Dex treatment.

## Atrophy- and slow fiber-associated genes are upregulated in fast-dominant muscles in Dex-treated LSD1-mKO mice

To gain functional insight into the role of LSD1 in muscle plasticity, we performed RNA-sequencing (RNA-seq) analysis of the Gas muscle after Dex treatment. We found 150 upregulated and 117 downregulated genes, respectively, in LSD1-mKO mice, compared with those in the WT (*Figure 2A*). Consistent with the phenotype, a group of genes associated with muscle atrophy were upregulated in LSD1-mKO mice (*Figure 2B* and *Supplementary file 1*). By using quantitative reverse transcription-polymerase chain reaction (qRT-PCR) analysis, we confirmed that the key atrophy mediators, including *Gadd45a*, *Gabarapl1*, and *Hspb7*, were significantly upregulated in the KO mice (*Figure 2C*; *Ebert et al., 2012*; *Schaaf et al., 2016*; *Tobin et al., 2016*; *Bullard et al., 2016*). In parallel, hypertrophy genes were downregulated in the LSD1-mKO Gas muscle (*Figure 2B* and *Figure 2—figure supplement 1A*). Furthermore, in the LSD1-mKO mice, the slow fiber genes, including *Sln* (encoding

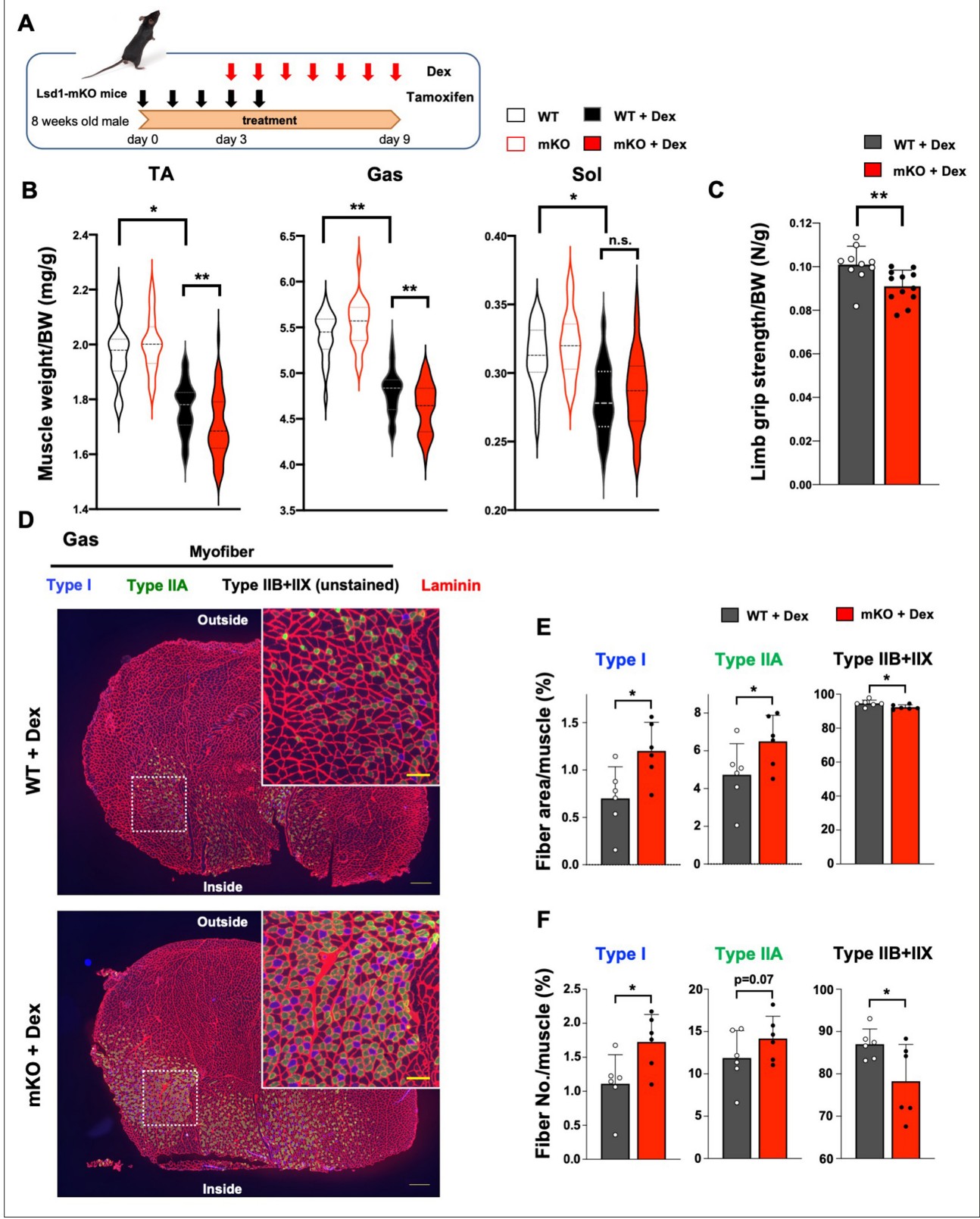

**Figure 1.** Muscle-specific lysine-specific demethylase-1 (LSD1)-knockout exacerbates dexamethasone (Dex)-induced fast muscle atrophy.
(**A**) Experimental design: LSD1 deletion was induced by tamoxifen injection for 5 days (days 0–4), and Dex was administered for 7 days (days 3–9) at 1 mg/kg. (**B**) Muscle weight normalized by the body weight. Tibialis anterior (TA), gastrocnemius (Gas), and soleus (Sol) muscles were obtained from Dex-treated wild type (WT) (n=40) and LSD1-mKO (n=44) mice. In the control set, WT (n=22) and mKO (n=22) mice that were not subjected to Dex

*Figure 1 continued on next page*

*Figure 1 continued*

treatment were sacrificed 1 month after the first tamoxifen administration. Box-plot elements (center line, median; box limits, upper and lower quartiles; whiskers, min to max). (C) All-limb grip strength of Dex-treated WT (n=10) and LSD1-mKO (n=12) mice. (D) Immunofluorescence staining of individual fiber types in Gas muscles of WT and LSD1-mKO mice after Dex treatment. Left: cross section of the whole tissue (scale bars, 300 µm). Right: magnified images of dotted squares in the images on the left (scale bars, 100 µm). Type I fiber (blue), type IIA fiber (green), and laminin (red) were stained. The fibers that were negative for types I and IIA were defined as type IIB+IIX. (E) Occupancy of each fiber type based on the cross-sectional area (WT, n=6; mKO, n=6). (F) Frequency of each muscle fiber type based on the fiber number (WT, n=6, mKO, n=6). Values are mean ± standard deviation (SD). *p<0.05, **p<0.01.

The online version of this article includes the following source data and figure supplement(s) for figure 1:

**Figure supplement 1.** Generation of lysine-specific demethylase-1 (LSD1)-mKO mice.

**Figure supplement 1—source data 1.** Original blots for *Figure 1—figure supplement 1C*.

**Figure supplement 2.** Effect of lysine-specific demethylase-1 (LSD1)-mKO at the postnatal developmental stage.

**Figure supplement 3.** Effect of dexamethasone (Dex) dosage on atrophic phenotype.

**Figure supplement 4.** Effects of dexamethasone (Dex) treatment in lysine-specific demethylase-1 (LSD1)-mKO mice.

**Figure supplement 5.** Effects of lysine-specific demethylase-1 (LSD1)-mKO on muscle atrophy- and hypertrophy-associated signaling pathways.

**Figure supplement 5—source data 1.** Original blots for *Figure 1—figure supplement 5*.

**Figure supplement 6.** Increase in the number of slow fibers in dexamethasone (Dex)-treated lysine-specific demethylase-1 (LSD1)-mKO mice.

**Figure supplement 7.** Fiber-type composition in dexamethasone (Dex)-treated lysine-specific demethylase-1 (LSD1)-mKO mice.

sarcolipin), a functional hallmark of oxidative fibers (*Maurya et al., 2018*), were significantly upregulated (*Figure 2B and D*). In contrast, fast fiber genes were downregulated in these mice. We also observed a similar expression pattern for fiber type-associated genes in the TA muscle (*Figure 2—figure supplement 1B*). These expression changes were not observed in the muscles that were not treated with Dex (*Figure 2—figure supplement 1C and D*). Intriguingly, RNA-seq analysis of the Sol muscle showed that the differentially regulated genes (WT vs. KO) were incompatible with those in the Gas muscle (*Figure 2E* and *Figure 2—figure supplement 2A and B*). In particular, a number of atrophy-associated genes were downregulated in the LSD1-mKO Sol, which was in clear contrast to the expression changes observed in the Gas (*Figure 2C* and *Figure 2—figure supplement 2C and D*). Taken together, these results indicate that LSD1 buffers Dex effects by repressing the atrophy- and slow fiber-associated transcriptome in fast-dominant muscle.

We next tested whether LSD1 has protective function in other atrophy models. For this, we administrated a pro-diabetic agent streptozotocin (STZ), which disrupts the function of pancreatic islet leading to fast-fiber atrophy in mice (*Figure 2—figure supplement 3A*; *O'Neill et al., 2019*). LSD1-mKO did not affect the muscle weight in STZ-treated mice (*Figure 2—figure supplement 3B and C*). Consistently, there were no major differences in the expression of atrophy and myofiber genes in STZ-treated WT and LSD1-mKO mice (*Figure 2—figure supplement 3D–E*). These results suggest that the LSD1 function depends on the source of atrophy-inducing stress.

## LSD1 regulates atrophy-associated genes by promoting the nuclear function of Foxk1

We previously reported that LSD1 inhibition led to an accumulation of tri-methylated histone H3 lysine 4 (H3K4me3) at the LSD1-bound promoters in differentiating myoblasts (*Anan et al., 2018*). To understand the mechanism through which LSD1 regulates the atrophy program, we evaluated the effects of LSD1 depletion on the H3K4 methylation status. We found that the H3K4me3 levels at the promoter regions of LSD1-mKO-suppressed genes were higher in the KO Gas muscle compared to those in the WT muscle (*Figure 3A*). In addition, we analyzed a previously published chromatin immunoprecipitation (ChIP)-seq dataset in mouse myoblasts (*Tosic et al., 2018*), and found that LSD1 bound to the promotor regions of the atrophy genes that were upregulated in LSD1-mKO Gas (*Figure 3—figure supplement 1A*).

To identify the transcriptional regulators that mediate the regulation of atrophy-associated genes by LSD1, we performed motif analyses on the promotor regions of upregulated genes in LSD1-mKO Gas and identified the binding motifs for Foxk1, PITX1, and NFATc1 (*Figure 3B*). In C2C12 cells, Foxk1 represses atrophy genes under nutrient-rich conditions and becomes sequestered from the nucleus

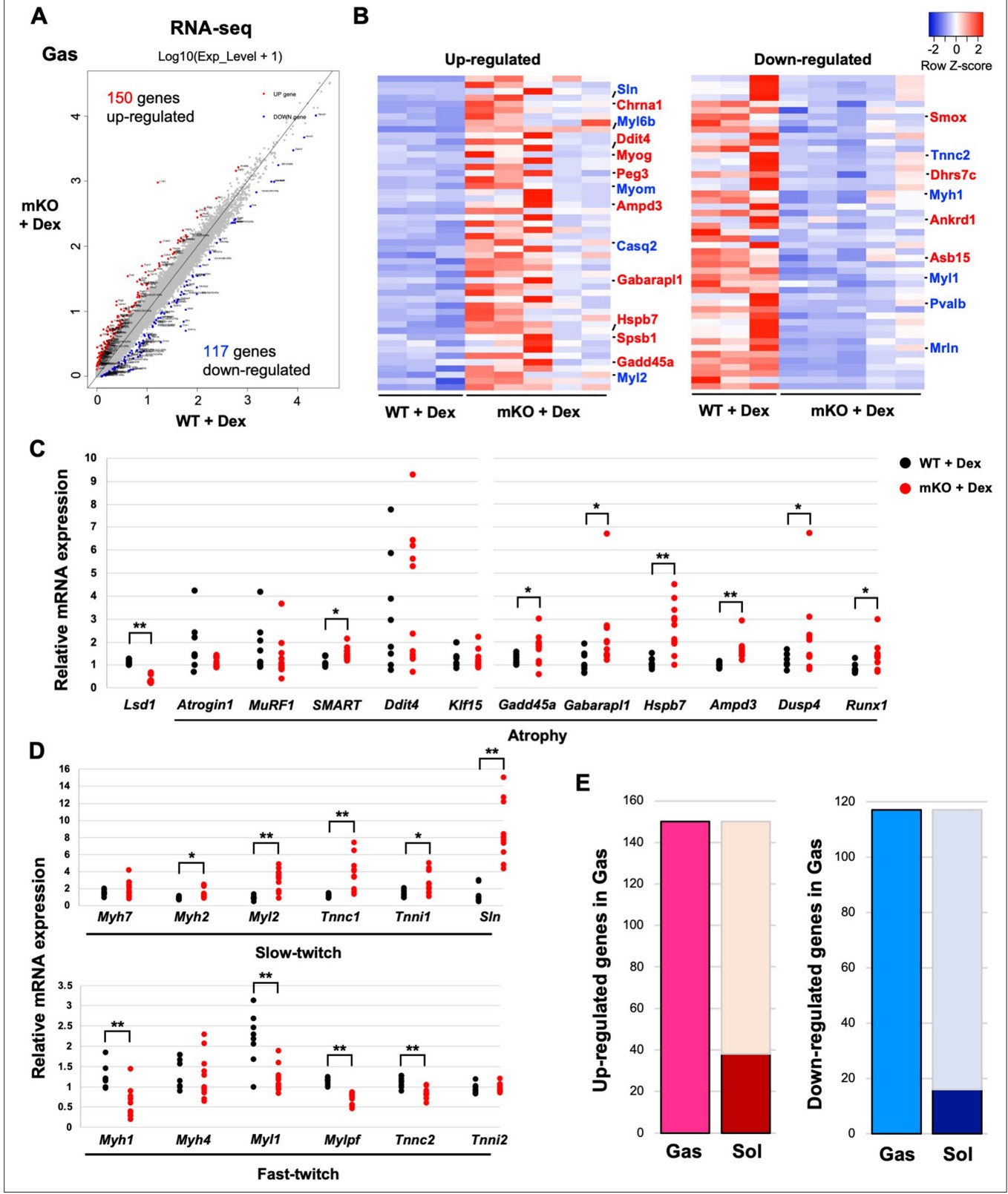

**Figure 2.** Transcriptome analysis reveals the upregulation of muscle atrophy and slow fiber genes in fast-dominant muscles in dexamethasone (Dex)-treated lysine-specific demethylase-1 (LSD1)-mKO mice. (**A**) Comparison of the transcriptome data from LSD1-mKO and wild type (WT) gastrocnemius (Gas) muscles. The red and blue dots indicate significantly up- and downregulated genes, respectively, in LSD1-mKO (FDR >0.05). (**B**) Top 50 differentially expressed genes ranked by the FDR Z-score. In the upregulated gene list, muscle atrophy- (red) and slow fiber-associated (blue) genes

*Figure 2 continued on next page*

*Figure 2 continued*

are highlighted. In the downregulated list, muscle hypertrophy- (red) and fast fiber-associated genes (blue) are highlighted. The complete gene list is provided in *Supplementary file 1*. (**C and D**) Quantitative reverse transcription-polymerase chain reaction (qRT-PCR) analysis of atrophy- (**C**) and fiber type-associated genes (**D**) in the Dex-treated LSD1-mKO Gas muscles (WT, n=8; mKO, n=11). Expression levels were normalized to those of the *36B4* gene and shown as fold differences against the WT. Full descriptions of gene symbols are provided in *Supplementary file 3*. (**E**) Comparison of the transcriptome profiles of Gas and soleus (Sol) muscles from Dex-treated LSD1-mKO mice. The dark red and blue bars indicate the fraction of genes that are commonly upregulated and downregulated, respectively, in the mKO Gas and Sol muscles. The numbers of genes that were up- or downregulated in mKO Gas are also indicated. Values are mean ± SD. *p<0.05, **p<0.01.

The online version of this article includes the following source data and figure supplement(s) for figure 2:

**Source data 1.** Lists of genes described in *Figure 2A*.

**Figure supplement 1.** Effects of lysine-specific demethylase-1 (LSD1)-mKO on the expression of atrophy-, hypertrophy-, and fiber type-associated genes.

**Figure supplement 2.** Transcriptome analysis of the soleus (Sol) muscle in lysine-specific demethylase-1 (LSD1)-mKO mice after dexamethasone (Dex) treatment.

**Figure supplement 2—source data 1.** Lists of genes described in *Figure 2—figure supplement 2A*.

**Figure supplement 3.** The role of lysine-specific demethylase-1 (LSD1) in streptozotocin (STZ)-induced muscle atrophy.

upon starvation (*Bowman et al., 2014*). In addition, the repressive function of Foxk1 is achieved through its interaction with Sin3A (*Bowman et al., 2014*), which also forms a complex with LSD1 (*Yang et al., 2018*). Of note, the expression of Foxk1 in the Gas and TA muscles was higher than that in the Sol muscle (*Figure 3C*), raising the possibility that Foxk1 mediates the muscle site-dependent function of LSD1. Immunofluorescence and western blot analyses in C2C12 myotubes showed that Dex exposure reduced the nuclear retention of Foxk1, which was further promoted by the addition of T-3775440, an LSD1 inhibitor (*Figure 3D* and *Figure 3—figure supplement 1B*). We also performed a co-immunoprecipitation (co-IP) experiment and found a physical interaction between Foxk1 and LSD1 in C2C12 myotubes (*Figure 3E*). In addition, both LSD1 and Foxk1 interacted with the common partner Sin3A (*Figure 3—figure supplement 1C*). Furthermore, we created Foxk1-KO C2C12 cells using a CRISPR Cas9 system and demonstrated that the key LSD1-targeted atrophy genes in these cells were upregulated after differentiation (*Figure 3F* and *Figure 3—figure supplement 2A*). Foxk1-KO C2C12 myotubes also showed upregulated expression of type I and IIA fiber genes, which was consistent with the results observed in the LSD1-mKO Gas muscle (*Figure 3F*). In these cells, the LSD1 inhibition did not further affect the expression of atrophy and myofiber genes (*Figure 3—figure supplement 2B*). ChIP experiment revealed that the occupancy of LSD1 at atrophy gene loci was diminished in the absence of Foxk1 (*Figure 3G* and *Figure 3—figure supplement 2C*). These data indicate that LSD1 largely depends on Foxk1 to regulate the atrophy genes. Moreover, the upregulation of atrophy genes was also observed in the C2C12 myotubes expressing Foxk1 shRNA (*Figure 3—figure supplement 2D and E*). Altogether, these results suggests that LSD1 and Foxk1 cooperatively regulate the atrophy transcriptome in the presence of Dex.

## Muscle-specific LSD1 depletion promotes the hypertrophy of Sol muscle and the muscle endurance after voluntary exercise

The differential effects of LSD1 depletion on the transcriptome in fast- and slow-dominant muscles under Dex administration motivated us to explore the physiological role of LSD1 in the slow-dominant muscles. For this purpose, we subjected the mice to voluntary wheel running (VWR). VWR augments muscle endurance capacity and induces transcriptomic changes in muscles in a site-dependent manner (*Legerlotz et al., 2008*; *Kim et al., 2020*; *Schmitt et al., 2020*). In particular, VWR has been shown to induce the hypertrophy of soleus in mice and rats (*Legerlotz et al., 2008*; *Momken et al., 2005*; *Konhilas et al., 2005*). To allow VWR, we kept LSD1-mKO and WT mice in a cage attached to a running wheel for over 1 month (*Figure 4A*). We confirmed that tamoxifen administration during the first 5 days of VWR sufficiently maintained an LSD1-deficient state until sacrifice at day 34 (*Figure 4A* and *Figure 4—figure supplement 1A*). The mice trained by VWR showed a prominent weight gain in the Sol muscle, which was further augmented by LSD1 depletion (*Figure 4B* and *Figure 4—figure supplement 1B*). Histologically, the area of type I muscle fibers significantly increased in the LSD1-KO Sol (*Figure 4C and D*, and *Figure 4—figure supplement 1C–E*). In a treadmill endurance test,

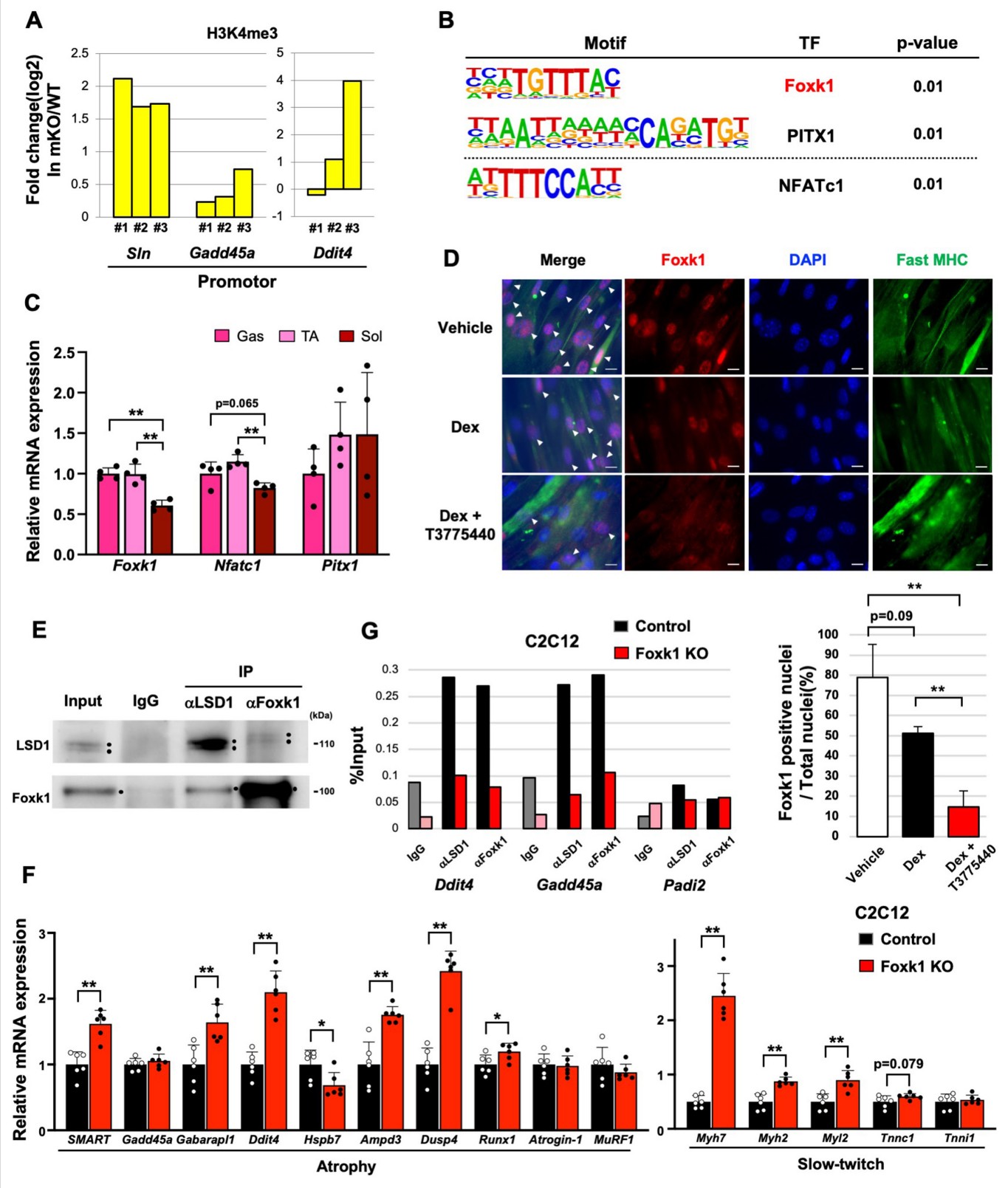

**Figure 3.** Lysine-specific demethylase-1 (LSD1) regulates the expression of atrophy-associated genes through the modulation of Foxk1. (**A**) Chromatin immunoprecipitation (ChIP)-qPCR analyses of H3K4me3 at *Sln*, *Gadd45a*, and *Ddit4* gene promoters in LSD1-mKO gastrocnemius (Gas) muscles after dexamethasone (Dex) treatment. The data are from three independent experiments. The enrichment levels of H3K4me3 were normalized to the input DNA and presented as fold differences in the LSD1-mKO against that of wild type (WT). (**B**) Motif analyses of the promoter regions of genes that were

*Figure 3 continued on next page*

*Figure 3 continued*

upregulated in Dex-treated LSD1-mKO Gas muscles. Analysis of the atrophy-associated genes identified Foxk1 and PITX1 motifs, while significant enrichment of NFATc1 motif was found when all the upregulated genes were scanned. (**C**) Expression of genes encoding the transcription factors (TFs) identified by the motif analyses. Quantitative reverse transcription-polymerase chain reaction (qRT-PCR) values (n=4) are shown as the fold differences, compared with those of the Gas muscles. (**D**) Effects of treatment with Dex and LSD1 inhibitor (T-3775440) on the expression of Foxk1 in C2C12 myotubes. Foxk1 and 5,6-diamidino-2-phenylindole (DAPI) co-staining revealed Foxk1-positive nuclei, which are indicated by arrowheads (scale bars, 20 μm). The graph shows the percentage of Foxk1-positive nuclei (n=3). (**E**) Co-immunoprecipitation of LSD1 and Foxk1. C2C12 myotubes were treated with insulin for 6 hr before harvest to enhance the nuclear retention of Foxk1. Input lane contains 10% amount of the whole-cell extract. (**F**) Effects of Foxk1-KO on the expression of atrophy- and slow fiber-associated genes in C2C12 myotubes (n=6). qRT-PCR values are shown as the fold differences, compared with those in the control-transfected cells (n=6). (**G**) ChIP-qPCR analyses of LSD1 and Foxk1 at Ddit4, Gadd45a, Padi2 (negative control) gene promoters in control and Foxk1-KO C2C12 myotubes. The enrichment levels of normal rabbit IgG, LSD1, and Foxk1 were normalized to the input DNA. Values are mean ± SD. *p<0.05, **p<0.01.

The online version of this article includes the following source data and figure supplement(s) for figure 3:

**Source data 1.** Original blots for *Figure 3E*.

**Figure supplement 1.** Lysine-specific demethylase-1 (LSD1) interacts with Foxk1 and increases the nuclear retention.

**Figure supplement 1—source data 1.** Original blots for *Figure 3—figure supplement 1B and C*.

**Figure supplement 2.** Foxk1 cooperates with lysine-specific demethylase-1 (LSD1) to control the expression of atrophy-associated genes.

**Figure supplement 2—source data 1.** Original blots for *Figure 3—figure supplement 2A and D*.

LSD1-mKO mice ran a longer distance and for a longer duration than the WT mice (*Figure 4E*), while the grip strength was comparable between the two genotypes (*Figure 4—figure supplement 1F*). Consistent with the results of the augmented endurance capacity, the Sol muscle in the KO mice showed enhanced succinate dehydrogenase (SDH) staining, indicating that the OXPHOS capacity of Sol was enhanced by the loss of LSD1 (*Figure 4F* and *Figure 4—figure supplement 1G*). The total running distance during the VWR was not influenced by genotype (*Figure 4—figure supplement 1H*), eliminating the possibility that the observed phenotypes were related to exercise intensity. Collectively, these data show that LSD1 depletion in the muscle promoted adaptation to prolonged physical activities by enhancing hypertrophy of the slow muscle.

## LSD1 depletion upregulates the expression of ERRγ and mitochondrial metabolism genes after voluntary exercise

To identify the genes regulated by LSD1 under VWR, we subjected the Sol muscle from trained WT and LSD1-mKO mice to RNA-seq analysis (*Figure 5A*). We performed a non-hierarchical clustering and identified a cluster of genes (cluster 6) that were upregulated in the LSD1-mKO. Consistent with the altered oxidative capacity, this cluster was associated with gene ontology terms, such as 'mitochondrion', 'mitochondrion organization', and 'oxidative phosphorylation' (*Figure 5B*). Of note, we found a significant upregulation of *Esrrg* gene, which encodes ERRγ, a TF that promotes oxidative metabolism in the muscle (*Figure 5C and D*, and *Supplementary file 2*; *Misra et al., 2017*; *Fan et al., 2018*; *Gan et al., 2013*; *Narkar et al., 2011*). In sedentary mice, the expression of *Esrrg* was not affected by LSD1 depletion (*Figure 5—figure supplement 1A*), suggesting a combinatorial effect of VWR and LSD1-mKO. The gene encoding PGC-1α, a pivotal transcriptional factor that promotes oxidative metabolism and endurance capacity (*Handschin et al., 2007*), was not affected in the LSD1-mKO mice (*Figure 5D*). Interestingly, the protein level of ERRγ was upregulated by VWR and further by LSD1 depletion, suggesting that the exercise-activated ERRγ function was potentiated in the LSD1-mKO muscle (*Figure 5E*). As we performed a motif analysis of upstream regions of the cluster 6 genes, we found a significant enrichment of the ERRγ binding motif (*Figure 5F*). qRT-PCR results confirmed that the known ERRγ targets as well as the genes harboring the ERRγ motif were upregulated in the LSD1-mKO Sol (*Figure 5G*; *Cho et al., 2013*; *Liang et al., 2016*). The expression of ERRγ and its target genes was not influenced by LSD1 depletion in the Gas and TA samples (*Figure 5—figure supplement 1B*).

To further test whether *Esrrg* expression is downregulated by LSD1, we made use of H9c2 rat cardiomyocytes, in which ERRγ is functionally involved in the differentiation (*Sakamoto et al., 2022*). Inhibition of LSD1 in H9c2 cells led to significant upregulation of *Esrrg* and its downstream Perm1 (*Figure 5—figure supplement 1C*), whereas these genes were downregulated in LSD1-overexpressing

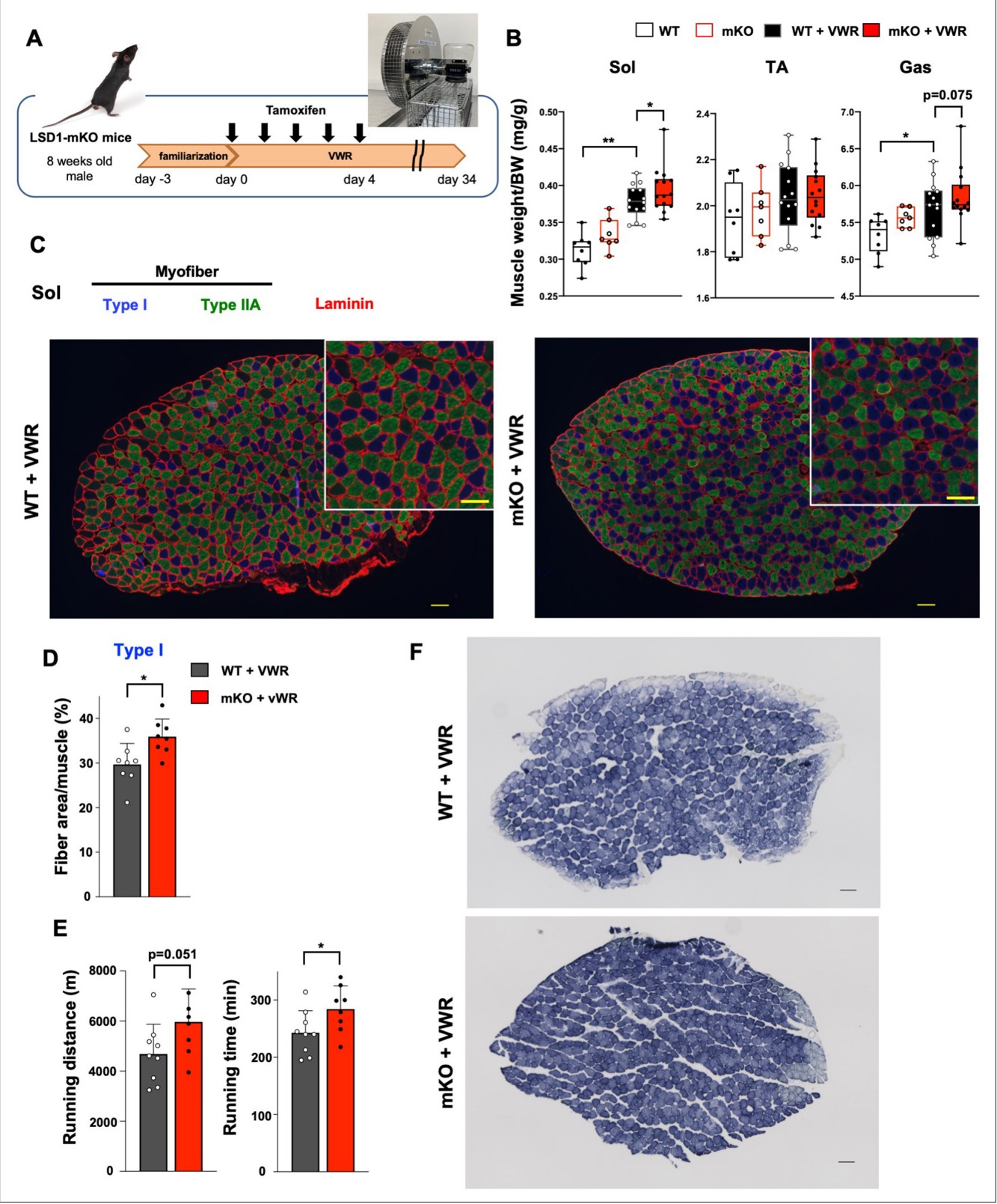

**Figure 4.** Lysine-specific demethylase-1 (LSD1)-mKO augments the effects of voluntary wheel running (VWR) by enhancing muscle endurance and increasing the number of oxidative fibers. (**A**) Experimental design: after familiarization with a cage equipped with a running wheel, LSD1 deletion was induced by tamoxifen injection (days 0–4). Mice were subjected to VWR in the same cage for 38 days (days –3–34). (**B**) Muscle weight normalized to body weight in sedentary wild type (WT) (n=8) and LSD1-mKO mice (n=7) as well as in VWR-trained WT (n=23) and LSD1-mKO mice (n=22).

*Figure 4 continued on next page*

*Figure 4 continued*

(**C**) Immunofluorescence staining of individual fiber types in the whole soleus (Sol) muscles from VWR-trained WT and LSD1-mKO mice. Type I fiber (blue), type IIA fiber (green), and laminin (red) were stained. Magnified images of the tissue sections (scale bars, 100 μm). Representative images (WT and mKO, n=8) are shown. (**D**) Occupancy of type I fiber based on the cross-sectional area (WT and mKO, n=8). (**E**) Treadmill performance of WT (n=9) and LSD1-mKO (n=8) mice after VWR. Running distance and time until exhaustion are shown. (**F**) Succinate dehydrogenase (SDH) staining of the Sol muscles from trained WT and LSD1-mKO mice. Scale bar, 100 μm.

The online version of this article includes the following figure supplement(s) for figure 4:

**Figure supplement 1.** Lysine-specific demethylase-1 (LSD1)-mKO increased the number of oxidative fibers in slow dominant muscle after voluntary wheel running (VWR) training.

H9c2 cells (*Figure 5—figure supplement 1D and E*). Collectively, these data suggest that LSD1 depletion enhanced oxidative metabolism and muscle endurance by promoting ERRγ activity in VWR-trained mice.

Finally, we tested whether aging affects the expression of LSD1 in the muscle. In mice, muscle weight declined with age, which was accompanied by the increased expression of atrophy genes (*Figure 6A and B*). In parallel, LSD1 expression was downregulated in aged mice both at RNA and protein levels (*Figure 6B and C*). Next, we analyzed the expression of LSD1 in individual muscle cells using single-cell RNA-seq data from the Tabula Muris Senis project (*Tabula Muris Consortium, 2020*). We found that *Lsd1* levels decline with age in both Myod1- and Pax7-positive myogenic cells (*Figure 6D*). In human muscle, *LSD1* expression showed a significant negative correlation with age (*Figure 6E*). These data, together with our observations in LSD1-mKO mice, imply that aging-associated loss of LSD1 confers the sensitivity against environmental stresses leading to qualitative and quantitative changes in the muscle.

## Discussion

Skeletal muscle exhibits notable plasticity in response to environmental cues, which can be beneficial or deleterious to health. In the current study, we demonstrated that the loss of LSD1 remarkably sensitized the skeletal muscle to Dex-induced atrophy and VWR-induced hypertrophy. Under these stressed conditions, LSD1 suppressed the catabolic pathways as well as the formation of type I fibers. Mechanistically, LSD1 cooperated with Foxk1 to repress protein catabolism genes, while repressing ERRγ and its downstream oxidative genes, depending on the stimulus and muscle location (*Figure 7*). Environmental adaptation involves transcriptional and epigenetic remodeling in response to environmental cues, which confer a phenotypic plasticity in cells (*Feil and Fraga, 2012*; *Peters et al., 2021*; *Simpson et al., 2011*; *Gilbert and Epel, 2009*; *Cavalli and Heard, 2019*). Epigenetic mechanisms do not only promote phenotypic alterations, but also serve as barriers that safeguard the identity and functions of cells under environmental fluctuations (*Hörmanseder, 2021*; *Flavahan et al., 2017*). While epigenetic factors that promote muscle adaptation have been reported (*Proserpio et al., 2013*; *Nayak et al., 2019*; *Luo et al., 2019*; *Moresi et al., 2010*; *Damal Villivalam et al., 2021*), factors that suppress environmental responses in the muscle were unknown. The current study reveals that LSD1 serves as an 'epigenetic barrier' that defines stress sensitivities in the muscle.

In the current study, we generated a muscle-specific LSD1-KO mice by crossing LSD1-floxed mice with ACTA1-CreERT2 mice. This enabled us to inactivate LSD1 in differentiating and differentiated muscle cells. We did not find major phenotypic changes in these mice without environmental insults. A previous report by *Tosic et al., 2018* demonstrated that the muscle-specific loss of LSD1 led to a decline in the regenerative capacity and the acquisition of brown adipocyte-like characteristics in the muscle cells. Tosic et al. made use of Pax7-cre to engineer *Lsd1* gene specifically in muscle satellite cells. Thus, it is likely that the difference in the engineering method accounts for the phenotypic divergence, and that LSD1 has a maturation stage-dependent function in the muscle cell lineage.

A cascade of signaling pathways and transcriptional regulators operate under GC exposure to induce atrophic gene expression (*Schakman et al., 2013*; *Kuo et al., 2013*). We demonstrated that LSD1 inhibition combined with Dex treatment led to impaired nuclear retention of Foxk1, resulting in the de-repression of atrophy genes. Previous reports showed that, under nutrient-rich conditions, the Akt-mTORC1 pathway promotes the nuclear localization of Foxk1 to repress autophagy and other catabolic genes (*Bowman et al., 2014*; *Sakaguchi et al., 2019*). Because GCs exert their pro-atrophic

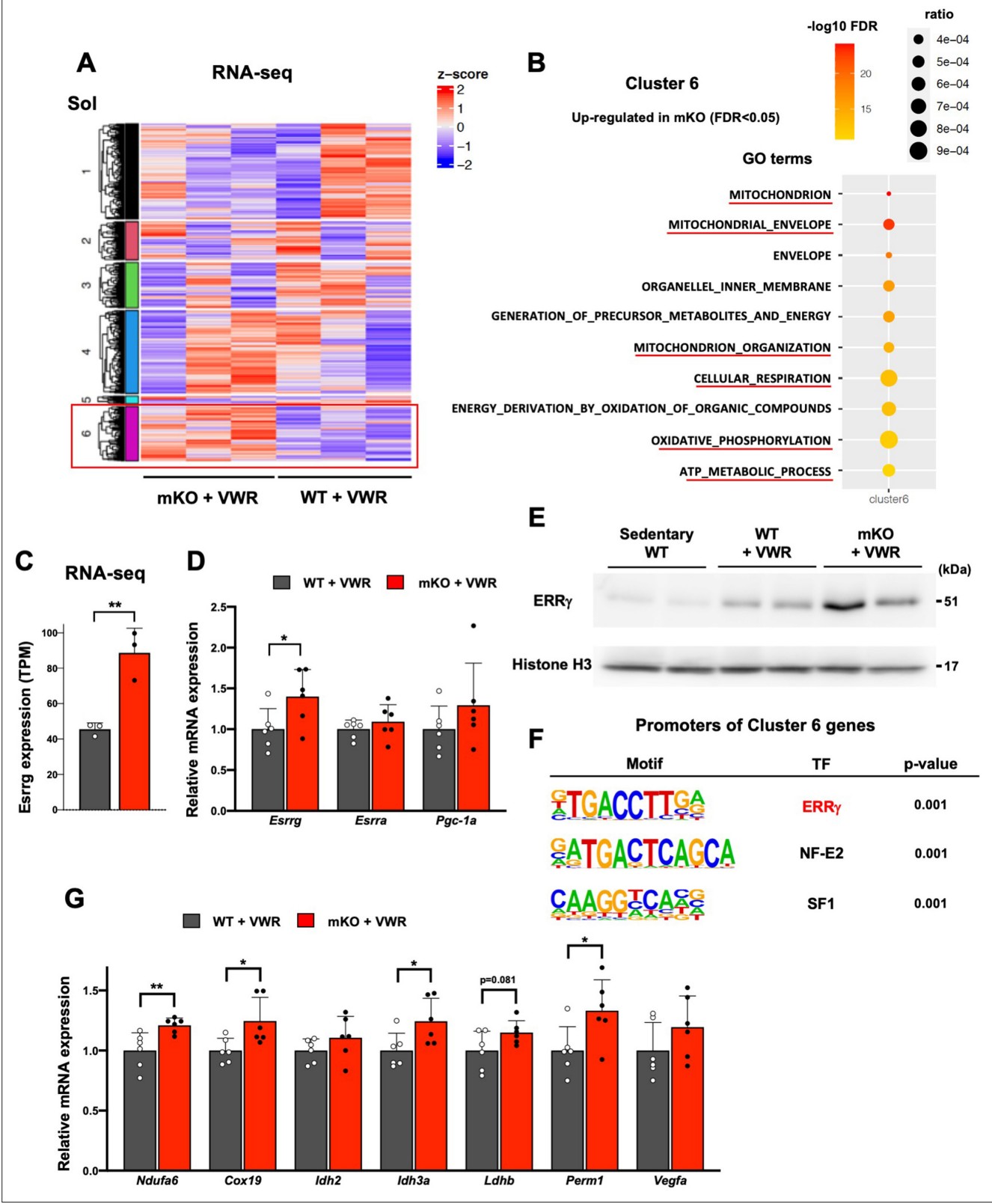

**Figure 5.** Lysine-specific demethylase-1 (LSD1)-mKO augments the effects of voluntary wheel running (VWR) by increasing the expression of ERRγ and its target genes. (**A**) RNA-seq analysis of wild type (WT) and LSD1-mKO soleus (Sol) muscles after voluntary wheel running (VWR) training. K-means clustering identified a group of genes that were upregulated in mKO (cluster 6). (**B**) RNA-seq analysis of WT and LSD1-mKO Sol muscles after VWR training. The genes upregulated in mKO (cluster 6) were subjected to a gene ontology analysis. (**C**) The expression of *Esrrg* in WT (n=3) and LSD1-mKO

*Figure 5 continued on next page*

*Figure 5 continued*

(n=3) Sol muscles after VWR training from RNA-seq data. (**D**) The expression of *Esrrg* and functionally related genes in WT (n=6) and LSD1-mKO (n=6) Sol muscles after VWR training. (**E**) The expression of ERRγ protein in Sol muscles from sedentary WT, trained WT, and trained LSD1-mKO mice. (**F**) Motif analyses of promoter regions of the cluster 6 genes. Values are mean ± SD. *p<0.05, **p<0.01. (**G**) Expression of ERRγ target genes in WT (n=6) and LSD1-mKO (n=6) Sol after VWR training. Quantitative reverse transcription-polymerase chain reaction (qRT-PCR) values are shown as the fold differences against the WT. Full descriptions of gene symbols are provided in the **Supplementary file 3**. Values are mean ± SD. *p<0.05, **p<0.01.

The online version of this article includes the following source data and figure supplement(s) for figure 5:

**Source data 1.** Original blots for *Figure 5E*.

**Figure supplement 1.** Expression of *Esrrg* and its target genes remain unaffected in lysine-specific demethylase-1 (LSD1)-mKO gastrocnemius (Gas) and tibialis anterior (TA) muscles after voluntary wheel running (VWR).

**Figure supplement 1—source data 1.** Original blots for *Figure 5—figure supplement 1D*.

effects partly through the inhibition of the Akt-mTORC1 axis (*Schakman et al., 2013*), this pathway may be involved in the nuclear retention of Foxk1. Thus, LSD1 counteracts GC signaling to facilitate the role of Foxk1, enabling optimal muscle adaptation to pro-atrophic stress. In addition, upon the sequestration of Foxk1 from the nucleus under starvation, Foxk1 target genes are replaced by Foxo3, which is known as a downstream effector of GR signaling (*Bowman et al., 2014*; *Cho et al., 2010*; *Sandri et al., 2004*). Thus, LSD1 depletion promotes the accessibility of Foxo3 or other Dex-induced TFs to atrophy genes in muscle cells.

Enhanced mitochondrial biogenesis and increased oxidative capacity are the key hallmarks of muscle adaptation to endurance exercise (*Qaisar et al., 2016*; *Egan and Zierath, 2013*). ERRγ is dominantly expressed in the slow-dominant muscles and induced in the fast muscles after exercise (*Gan et al., 2013*; *Narkar et al., 2011*). Forced expression of ERRγ in the muscle induces the formation of type I fibers and promotes endurance capacity, while the loss of ERRγ is associated with impaired oxidative metabolism and muscle endurance (*Misra et al., 2017*; *Narkar et al., 2011*; *Rangwala et al., 2010*). Moreover, comprehensive epigenomic analysis in the mouse muscles identified ERRγ as the causal TF for exercise-induced epigenomic reprogramming (*Ramachandran et al., 2019*). This evidence indicates that ERRγ is a key mediator of the metabolic and physical adaptation to exercise. Previous studies have primarily focused on the function of ERRγ in the fast-dominant muscles, and hence, its physiological relevance in the slow muscles has not been characterized. In this study, we found that LSD1 depletion promotes the adaptive response to VWR, with increased expression of ERRγ and its downstream mitochondrial metabolism genes in the Sol. We also found that VWR upregulated the protein expression of ERRγ in the Sol, and hence, our data collectively indicates that LSD1 limits the adaptive response in the slow muscle, particularly the type I fiber, by fine-tuning the exercise-induced expression of ERRγ.

Fiber-type composition in the skeletal muscle is subject to remodeling under diverse environmental stresses. Previous reports have demonstrated that Dex treatment increases the relative occupancy of type I fibers in fast-dominant muscles due to the shrinking of type IIB/X fibers but does not directly affect the growth or formation of type I fiber (*Ciciliot et al., 2013*; *Shimizu et al., 2015*). Notably, we observed an increase in the number of enlarged type I fibers in the LSD1-mKO Gas muscle. Because there were no obvious de novo fiber formation under our experimental setting (*Figure 1—figure supplement 7C*), it was likely that the preexisting fibers were affected by LSD1 depletion. One possible explanation is that the slow fibers in the LSD1-mKO mice became resistant to Dex. However, previous reports have shown that the type I fibers are unresponsive to GC-induced shrinking (*Ciciliot et al., 2013*; *Shimizu et al., 2015*), suggesting that this fiber type in the Gas was already resistant to Dex, regardless of the LSD1 genotype. A more plausible scenario is that type II fibers were converted to type I fibers. This possibility is backed by our previous finding that LSD1 binds to and represses a number of slow fiber-associated genes (e.g., Myh7) in C2C12 myotubes that express fast muscle-like phenotypes (*Anan et al., 2018*). Thus, the loss of LSD1 compromised the fast fiber identity, which in turn made the muscle cells permissive to the fiber-type conversion induced by the Dex treatment.

In our VWR study, we found that the loss of LSD1 led to an increase in the type I fiber area but not the fiber number, suggesting that these fibers were selectively enlarged. It has also been reported that VWR induces the hypertrophy of Sol with prominent effects on type IIA fibers but not on type I fibers (*Momken et al., 2005*; *Pellegrino et al., 2005*). Thus, our data indicate that LSD1 loss sensitized type I fibers to exercise-induced stress. Because LSD1-KO did not upregulate the slow fiber-associated

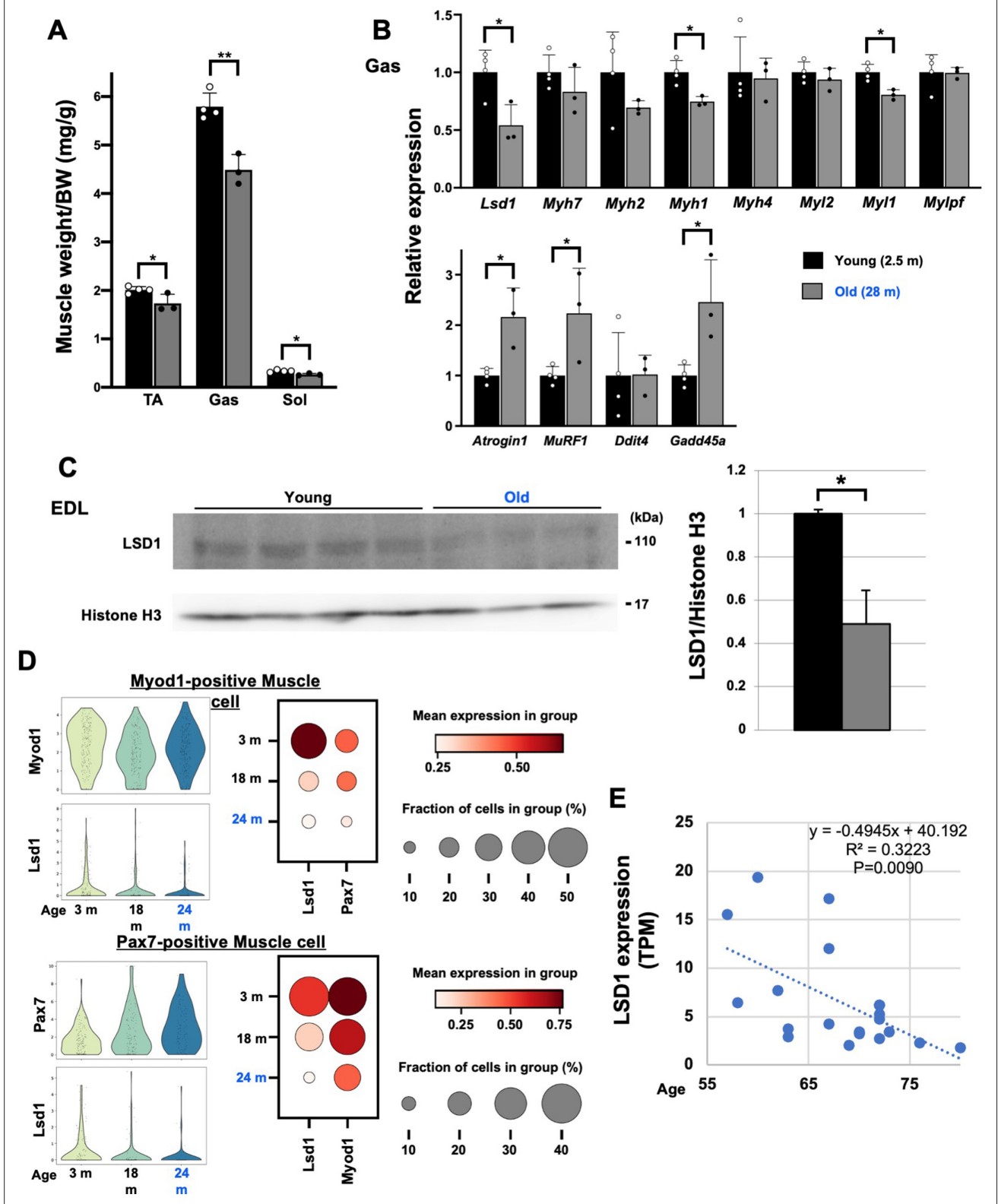

**Figure 6.** Lysine-specific demethylase-1 (LSD1) expression is decreased in the aged muscles in mice and humans. (**A**) Muscle weight in young (2.5 months of age, n=4) and old (28 months of age, n=3) male C57BL/6J mice. (**B**) The expression of fiber type-specific genes and atrophy-associated genes in the gastrocnemius (Gas) muscles in young and old mice. Quantitative reverse transcription-polymerase chain reaction (qRT-PCR) values are shown as fold differences against the young mice. (**C**) Expression of LSD1 protein in the EDL muscles from the young and old mice. Band densities

*Figure 6 continued on next page*

*Figure 6 continued*

were quantified by densitometry and normalized to histone H3. Values are shown as fold differences against the young. (**D**) Decreased expression of *Lsd1* in aged muscles. Using single-cell RNA-seq data (flow cytometry) published by the Tabula Muris Senis (*Tabula Muris Consortium, 2020*), *Lsd1* expression was analyzed in Myod1-positive and Pax7-positive muscle cells from young to old mice (3–24 months of age). (**E**) The expression of *LSD1* in skeletal muscles of 55- to 80-year-old men and women. RNA-seq data by *Tumasian et al., 2021*, was analyzed. Values are mean ± SD. $R^2$: coefficient of determination. *p<0.05, **p<0.01.

The online version of this article includes the following source data for figure 6:

**Source data 1.** Original blots for *Figure 6C*.

genes, such as *Myh7* and *Myl2* (data not shown), LSD1 possibly suppressed the synthesis of type I fiber indirectly by negatively regulating the translation or stability of fast fiber proteins. In a previous report, the forced expression of oxidative metabolism genes in the muscle in mice enhanced type I fiber formation (*Hénique et al., 2015*). Thus, it is possible that the enhanced oxidative capacity in the trained LSD1-KO Sol triggered the type I fiber program. Collectively, our current study highlights the important contribution of LSD1 to the maintenance of fiber-type identity in the skeletal muscle under environmental fluctuations.

Our findings have several important implications from the biomedical standpoint. Sarcopenia is aging-associated muscle loss (*Cruz-Jentoft and Sayer, 2019*) and is characterized by a fast fiber-selective loss with the possible involvement of elevated GCs in aged individuals (*Ciciliot et al., 2013*; *Clegg and Hassan-Smith, 2018*). We found that LSD1 expression in mouse and human muscles declined with age and showed negative correlation with the expression of the atrophy genes (*Figure 6*). Revitalizing LSD1 activity using chemicals or dietary factors (e.g., riboflavin) may facilitate muscle maintenance under aging-associated stresses. Moreover, the inhibition of LSD1 in combination with aerobic exercise may be beneficial for the prevention and management of disuse-induced atrophy, in which the oxidative capacity in the slow muscles is impaired (*Ciciliot et al., 2013*; *Ji and Yeo, 2019*). The modulation of LSD1 activity in different contexts will provide new strategies for improving muscle health.

## Methods
### Reagents and antibodies

Dex and insulin (from bovine pancreas) were purchased from Sigma-Aldrich, and STZ was from Wako. An LSD1 inhibitor, T-3775440 hydrochloride, was from MedChemExpress. The primary antibodies used for western blot, co-IP, immunohistochemistry, and immunocytochemistry experiments were as follows: anti-LSD1 (Abcam, ab17721), anti-FOXK1 (Abcam, ab18196), anti-ERRγ (Abcam, ab12893), anti-MHC type I (Developmental Studies Hybridoma Bank [DHSB], BA-F8), anti-MHC type IIA (DHSB, SC-71), anti-Laminin (Sigma-Aldrich, L9393), anti-Akt (Cell Signaling Technology [CST], #9272), anti-Phospho-Akt (Ser473) (CST, #9271), anti-4E-BP1 (CST, #9454), anti-Phospho-4E-BP1 (Thr37/46) (CST, #2855), anti-SQSTM1(p62) (Santa Cruz, sc-28359), anti-GAPDH (Santa Cruz, sc-25778), anti-LC3(MBL, PM036), and anti-Sin3A (Santa Cruz, sc-994). The secondary antibodies used were as follows: anti-mouse IgG-horseradish peroxidase (GE Healthcare, NA931V), anti-rabbit IgG-horseradish peroxidase (GE Healthcare, NA934V or CST, #7074), Alexa Fluor 488 anti-Mouse IgG1 (Thermo Fisher, A21121), Alexa Fluor 350 anti-Mouse IgG2b (Thermo Fisher, A21140), and Cy3 anti-Rabbit IgG (Jackson ImmunoResearch, 711-165-152). The antibodies used for ChIP experiments were anti-tri-methylated histone H3K4 (Millipore, 07-473), anti-pan histone H3 (Abcam, ab1791), and normal rabbit IgG (Santa Cruz, sc-2027).

### Cell culture

C2C12 mouse myoblasts were obtained from RIKEN BRC Cell Bank and maintained in Dulbecco's Modified Eagle's Medium (DMEM, Sigma-Aldrich) supplemented with 10% (v/v) heat-inactivated fetal bovine serum (FBS) at 37°C and 5% $CO_2$. For myogenic induction, subconfluent C2C12 myoblasts were cultured in DMEM with 2% (v/v) horse serum, and the medium was changed every other day. For immunofluorescence study, C2C12 myotubes at differentiation day 6 were treated with 10 nM T3775440 hydrochloride or phosphate buffered saline (PBS), and then at day 8, the myotubes were

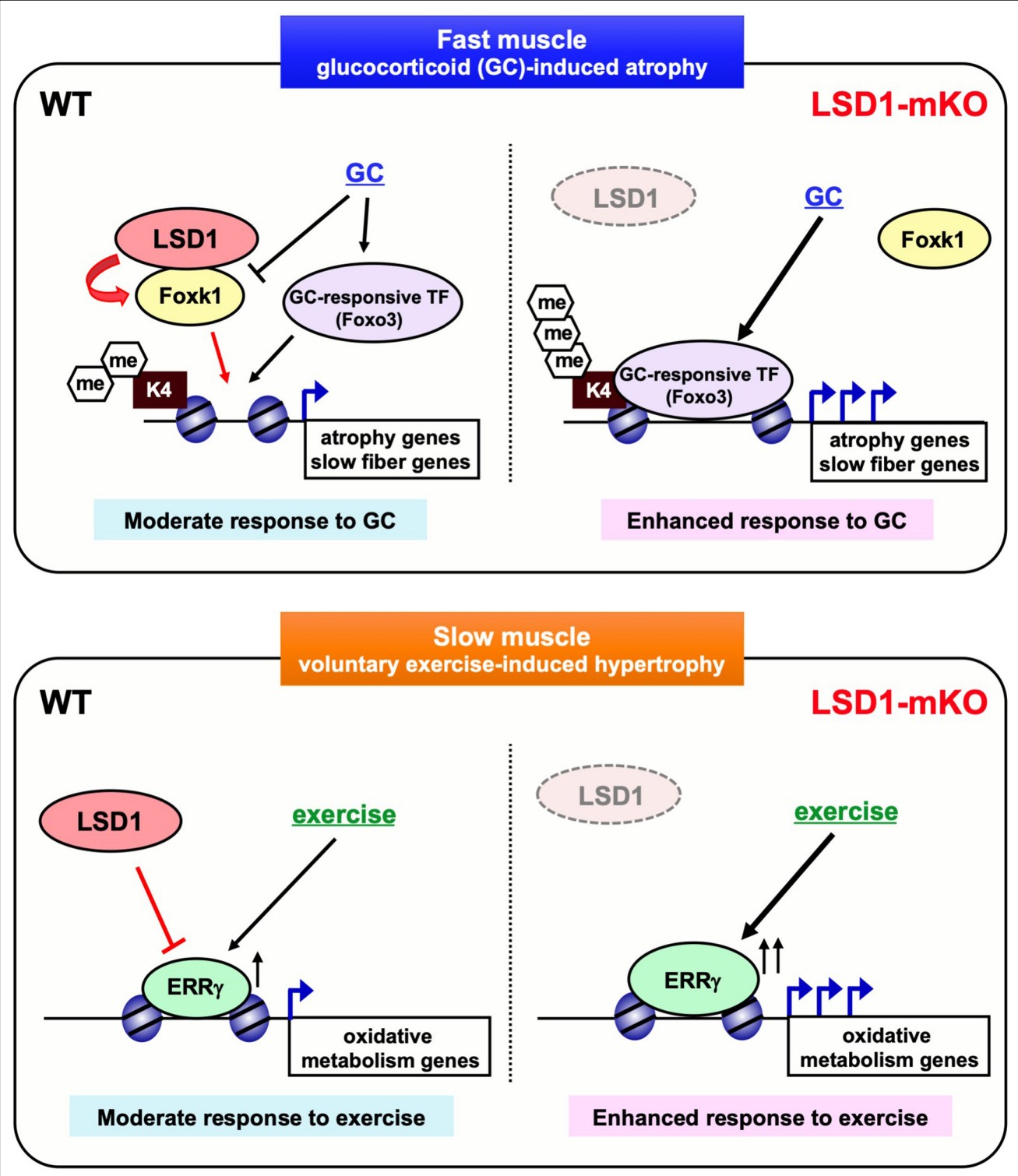

**Figure 7.** Lysine-specific demethylase-1 (LSD1) serves as an 'epigenetic barrier' that defines stress sensitivities in the skeletal muscle. LSD1 attenuates glucocorticoid (GC)-induced atrophy in the fast fiber-dominant muscles, in collaboration with Foxk1, an anti-autophagic transcription factor. On the other hand, LSD1 attenuates endurance exercise-induced hypertrophy in the slow fiber-dominant muscles, by inhibiting ERRγ, a transcription factor that promotes oxidative metabolism genes. The loss of LSD1 remarkably sensitized the muscles to GC and endurance exercise.

treated with 1 μM Dex or ethanol for 10 hr immediately before fixation. For co-IP experiments, C2C12 myoblasts were allowed to differentiate for 8 days, followed by treatment with 1 μM insulin for 6 hr.

For CRISPR-Cas9-based genome editing, either Control Double Nickase Plasmid (Santa Cruz Biotechnology, sc-437281) or FOXK1 Double Nickase Plasmid (Santa Cruz Biotechnology, sc421682-NIC) were transfected into C2C12 cells using a Fugene 6 reagent (Promega) according to the manufacturer's instructions. The cells were subjected to puromycin selection to establish stable lines.

H9c2 rat cardiomyocytes (ATCC) were cultured in DMEM supplemented with 10% FBS at 37°C and 5% $CO_2$. For differentiation, cells were cultured in DMEM with 1% FBS for 7 days.

Identities of the cell lines were confirmed by gene expression profiles and morphological changes during the differentiation. In addition, C2C12 (mouse) and H9c2 (rat) cells can be distinguished by using species-specific primers. Only the cells of low passage no. were used to ensure the phenotypic consistencies. These cell lines were tested negative for mycoplasma.

## Animal studies

Animal experiments were approved by the Animal Care and Use Committee of Kumamoto University (A2019-098), and conducted in accordance with its guidelines. Lsd1$^{flox/flox}$; $^{TgACTA1-CreERT2/+}$ conditional knockout mice (LSD1-mKO mice) were generated by crossing Lsd1-floxed mice (B6.129-Kdm1atm1.1Sho/J, Jackson Laboratories) and knock-in mice expressing a CreERT2 recombinase under the control of human ACTA1 gene promoter (Tg(ACTA1-cre/Esr1*)2Kesr/J, Jackson Laboratories). To induce Cre expression, 20 mg/mL tamoxifen in corn oil was administered intraperitoneally to 8-week-old or postnatal day 11 male LSD1-mKO or WT (Lsd1-floxed, without CreERT2) mice at a dose of 75 mg/kg for 5 consecutive days. After 1 week, RNA and protein were isolated from the tissues for qRT-PCR and western blot analyses, respectively, to evaluate the degree of LSD1 knockout (*Figure 1—figure supplements 1 and 2*).

To determine the effect of Dex treatment, Dex at 1 or 5 mg/kg or vehicle (PBS) was intraperitoneally administered to 8-week-old male mice for 7 consecutive days. These mice were either used for muscle function tests or sacrificed for tissue analysis after 6 hr fasting.

To evaluate the effect of insulin deficiency, STZ was dissolved in 0.1 M sodium citrate buffer (pH 4.5) to produce a concentration of 10 mg/mL. Two-hundred mg/kg STZ was administered intraperitoneally to 8-week-old male LSD1-mKO or WT mice on the fifth day of 5 consecutive days of tamoxifen administration. Insulin deficiency was confirmed 4 days later with a blood glucose reading of >350 mg/dL. After 2 weeks of STZ injection, RNA was isolated from the excised tissues for qRT-PCR.

## Voluntary wheel running

For VWR test, 8-week-old control or LSD1-mKO mice were single-housed in cages with free access to a vertical running wheel (Shinano Seisakusho, Tokyo, Japan) for 38 days. After 3 days of familiarization, LSD1-KO was induced by tamoxifen injection for 5 days, consecutively, and the running distance was measured for 30 subsequent days. Before being subjected to muscle function test and tissue analysis, the mice were kept away from the running wheel and subjected to fasting for 6 hr. Sedentary controls were single-housed in similar cages but without access to a running wheel.

## Muscle function tests

To assess muscle endurance, the mice were subjected to a low-intensity, run-to-exhaustion protocol on a motorized treadmill (Muromachi Kikai, Tokyo, Japan) (*Fujita et al., 2018*). First, the mice were allowed to familiarize with the treadmill for 10 min at 10 m/min for 2 days. The next day, the mice were placed on a treadmill moving at 10 m/min for 30 min, 11 m/min for 15 min, and 12 m/min for 15 min with a 15° incline. Finally, the speed was escalated with an increment of 1 m/min every 10 min until the mouse exhibited exhaustion. The timepoint at which the mice sat on the shock grid at the back of the treadmill for longer than 5 s was set as the endpoint.

To assess muscle strength, whole-limb grip strength was measured using a Grip Strength Meter (Columbus Instruments, Columbus, OH, USA) for mice, as previously described (*Fujita et al., 2018*). Peak tension (in newtons) was recorded when the mice released the grip. Two sets of 10 successive measurements were performed for each mouse and the mean maximum strength in the second set was used for data analysis.

## Histological analysis

Gas and Sol muscles were dissected and immediately frozen in liquid N2-cooled isopentane. Cryo-sections (10 μm) were prepared using a cryostat (Leica). The sections were dried at 25°C and stored at –80°C.

For immunohistochemical analysis, the sections were fixed in acetone for 5 min at 4°C, blocked in PBS containing 3% goat serum, and 0.1% bovine serum albumin (BSA) for 2 hr at RT and incubated overnight at 4°C with a mixture of the primary antibodies (αMYH type I, αMYH type IIA, and αLaminin) in Can Get Signal immunostain Solution A (TOYOBO). After three 10 min washes in 0.1% BSA/PBS, the sections were incubated for 1 hr with a mixture of three secondary antibodies against mouse IgG2b (Alexa 350), mouse IgG1 (Alexa 488), and rabbit IgG (Cy3) in Can Get Signal immunostain Solution A, followed by three 10 min washes in 0.1% BSA/PBS. The sections were then mounted and sealed with nail polish.

The sections were imaged with a KEYENCE BZ-X800 microscope and an Olympus BX51 wide-field microscope. The cross-sectional area and numbers of type I, type IIA, and IIB+IIX fibers were determined by analyzing individual fibers that were visualized by laminin staining. Quantification was done using a BZ-X Analyzer software (KEYENCE). Example images used for detecting individual fibers are shown in *Figure 1—figure supplement 6B*.

SDH staining was performed by exposing the cryosections to a mixture of 130 mM succinate, 1.5 mM nitroblue tetrazolium, 0.2 M $NaH_2PO_4$, and 0.2 M $Na_2HPO_4$ for 30 min at 37°C. The sections were then washed once with 70% and 95% ethanol and twice with 99.5% ethanol. Next, after washing with xylene, the sections were dried and mounted in malinol (Muto Pure Chemicals, Tokyo, Japan).

## Western blot and co-IP analyses

To prepare total cell lysates, the cells were collected and suspended in a sample buffer (0.1 M Tris-HCl, pH 6.8, 4% sodium dodecyl sulfate [SDS], 0.1 M dithiothreitol, 20% glycerol, and 0.2% bromophenol blue). To prepare tissue lysates, the frozen tissues were crashed in RIPA buffer (50 mM Tris-HCl, 150 mM NaCl, 2 mM EDTA, 1% NP-40, 0.1% SDS, and 0.5% sodium deoxycholate) by FastPrep-24 5G (MP Biomedicals, Santa Ana, CA, USA) at 6.0 m/s for 40 s with CoolPrep mode, then suspended in the sample buffer. Following sonication and centrifugation at 16,710 × $g$ for 5 min at 4°C, the supernatant was collected and used for western blotting. Protein samples were electrophoresed on an SDS polyacrylamide gel and then transferred to a nitrocellulose membrane (Amersham Protran Premium; GE Healthcare, Waukesha, WI, USA) using a semidry method. After blocking for 1 hr using 5% skim milk or 5% BSA dissolved in PBS containing 0.3% Tween 20, the membrane was incubated overnight at 4°C with primary antibodies in Can Get Signal solution (Toyobo, Osaka, Japan). The secondary antibodies used were anti-mouse IgG and anti-rabbit IgG-horseradish peroxidase. The blots were incubated for 1 min with Western Lightning Plus-ECL solution (PerkinElmer, Waltham, MA, USA) and visualized using Image-Quant LAS4000 Mini (GE Healthcare). For quantification, the band densities were determined using ImageJ (National Institutes of Health, Bethesda, MD, USA) software.

For co-IP experiments, the cells were lysed in an IP buffer (50 mM Tris-HCl, pH 8.0, 5 mM EDTA, 150 mM NaCl, 0.5% NP-40, and 0.5% Triton X-100) and incubated with specific antibodies, followed by pulldown using Dynabeads protein A/G (Life Technologies) for subsequent western blot analysis.

## Separation of nuclear and cytoplasmic fractions

Separation of nuclear and cytoplasmic fractions of C2C12 myotubes was performed as previously described (*Kim et al., 2010*). Briefly, cells were washed twice with cold PBS and harvested with a scraper, and then mixed with nuclear extraction buffer (20 mM HEPES, pH 7.5, containing 150 mM NaCl, 10% glycerol, 1 mM EDTA, 0.5% Triton X-100, 1 mM $Na_3VO_4$, 10 mM NaF, 1 mM PMSF, and 0.01% of protease inhibitor cocktail) and incubated for 30 min on ice. The cell suspension was passed through a 23-gauge needle 30 times and then centrifuged. The pellet and supernatant from the centrifugation are referred to as the nuclear and cytoplasmic fractions, respectively. The nuclear fraction was washed three times with nuclear extraction buffer and then suspended in the sample buffer.

## Quantitative RT-PCR

Total RNA was extracted from cells or frozen mouse tissues using a TRIzol reagent (Invitrogen). cDNA samples were produced using the ReverTra Ace qRT-PCR Kit (TOYOBO). qPCR-RT was performed by

the SYBR green method using a Thunderbird qPCR Mix (TOYOBO) and a StepOne Plus Sequence Detector (Applied Biosciences). Data are presented as the mean ± standard deviation (SD). Fold differences among groups were calculated by the ΔΔCt method. 36B4 (Rplp0) gene was used as an internal control. The primers are listed in *Supplementary file 3*.

## Poly (A) RNA-seq analysis

Total RNA was extracted from the Gas and Sol muscles using a Trizol reagent. mRNA was purified using a NEBNext Poly(A) mRNA Magnetic Isolation Module (New England Biolabs [NEB], Ipswich, MA, USA), and the sequencing libraries were synthesized using a NEBNext Ultra II RNA Library Prep Kit for Illumina (NEB). Sequencing was performed using a NextSeq 500 Sequencer (Illumina, San Diego, CA, USA) with 75 bp single-end reads. The resulting reads were aligned to the mm10 reference genome by using STAR (*Dobin et al., 2013*). For the dataset from Dex administration experiment, Edge R was used for the quantification of gene expression changes (*Robinson et al., 2010*), in which a false discovery rate cut-off of 0.05 was used to determine differentially expressed genes between WT and KO mice. Heatmaps were generated by using Heatmapper (*Babicki et al., 2016*). For the VWR dataset, DESeq2 was used for the quantification of gene expression changes (*Love et al., 2014*; *Takase et al., 2019*). The top 1000 differentially expressed genes between WT and KO mice were extracted from the genes with mean transcripts per million (TPM) >1 by using median absolute deviation (MAD). K-means clusters and heatmaps were generated by using ComplexHeatmap (*Gu et al., 2016*). Gene Ontology analysis was performed by using the Molecular Signatures Database (MSigDB) v7.4 (*Subramanian et al., 2005*).

Motif enrichment at the promoter regions was examined using the findMotifs.pl module in HOMER (V4.9.1) (*Heinz et al., 2010*). The parameters used for the Dex test were 'start –300, end +50, motif length 8, 10, 12', whereas those for the VWR test were 'start –2000, end +2000, motif length 10, 12'. For the Dex administration dataset, the LSD1-dependent atrophy genes and all upregulated genes in the LSD1-KO mice were subjected to motif analysis.

## ChIP-qPCR

To prepare a ChIP sample, frozen Gas muscles from two mice were minced in cooled PBS by using FastPrep-24 5G (MP Biomedicals), and the lysate was crosslinked with 1% formaldehyde. Next, to isolate the nuclei, tissue lysates were homogenized in cell lysis buffer (5 mM PIPES, 85 mM KCl, and 0.5% NP-40) with a Dounce homogenizer using a loose pestle. The isolated nuclei were resuspended in SDS-lysis buffer (1% SDS, 10 mM EDTA, and 50 mM Tris-HCl) and sonicated in two steps (*Kohrogi et al., 2021*): first with a probe-type sonifier (Branson) and second with a water bath sonifier, Picoruptor (Diagenode). For the C2C12 ChIP sample preparation, cells were dual crosslinked with 1% formaldehyde and disuccinimidyl glutarate (Thermo Fisher Scientific) to increase the stability of protein-DNA complexes. Following cell lysis, cells were treated with 4 U of Micrococcal nuclease per $1 \times 10^6$ nuclei at 37°C for 20 min, then sonicated by Bioruptor (Diagenode) for 10 min. Chromatin fragments were incubated at 4°C overnight with appropriate antibodies, followed by a pulldown using protein A/G-conjugated agarose beads. Purified DNAs were subjected to qPCR using the primer sets listed in *Supplementary file 4*.

### Immunocytochemistry

The cells were fixed with 4% paraformaldehyde in PBS for 15 min at 25°C. The cells were washed three times with PBS for 5 min and then permeabilized with PBS containing 0.5% Triton X-100 for 5 min on ice. The cells were blocked in 0.5% BSA/PBS at RT and incubated overnight at 4°C with a mixture of primary antibodies in 0.2% BSA/PBS. After three 10 min washes in 0.2% BSA/PBS, the cells were incubated for 1 hr with a mixture of secondary antibodies against mouse IgG1 (Alexa 488) and rabbit IgG (Cy3) in 0.2% BSA/PBS, followed by three washes in PBS. The cells were then mounted and sealed with nail polish. DNA was counterstained with 1 µg/mL 5,6-diamidino-2-phenylindole. Quantitative analysis was performed with BZ-X Analyzer software (KEYENCE).

### Construction of lentiviral expression vectors and their transduction

The plasmids for constructing and packaging lentiviral vectors were obtained from the RIKEN Bioresource Center (https://dna.brc.riken.jp/en/rvden/cfmen). Double-stranded DNA harboring an shRNA

sequence for FOXK1 was inserted into an entry vector, pENTR4-H1. Using the Gateway recombination method, an shRNA cassette was cloned into CS-RfA-EVBsd, a lentiviral plasmid. To produce lentivirus particles, packaging (pCAG-HIVgp) and envelope (pCMV-VSV-G-RSV-Rev) plasmids were co-transfected into Lenti-X 293T cells (Takara Bio) using Fugene 6 reagent (Promega). The culture supernatant was centrifuged, filtered, and stored at –80°C until use. C2C12 cells were infected with CS-RfA-EVBsd shFoxk1 (shFoxk1) or CS-RfA-EVBsd shGL3 (shGL3) (control shRNA targeting firefly luciferase gene) viruses, and then subjected to blasticidin S selection to establish stable lines. The short hairpin RNA target sequences were as follows: shFoxk1, 5'-ATCCAGTTCACATCGCTAT-3'; shGL3, 5'- CTTACGCT GAGTACTTCGA -3'.

For the forced expression of LSD1 in H9c2 cells, we constructed a lentiviral vector harboring human LSD1 cDNA under control of human elongation factor-1α (EF-1α) promoter. For this, we cloned FLAGx3-hLSD1 into a lentiviral plasmid CSII-EF-MCS (RIKEN) together with blasticidin resistance gene. After transduction, cells were selected with 10 µg/mL blasticidin S before use.

## Tabula Muris Senis

Single cell RNA-seq data from mice of different ages (3, 18, and 24 m) were obtained from the Tabula Muris Senis Project (*Tabula Muris Consortium, 2020*). FACS-sorted single-cell data from TA muscle (designated as limb muscle) were downloaded at GitHub (https://github.com/czbiohub/tabula-muris-senis, copy archived at swh:1:rev:18219fd8d2acaf9c8d46befbdfb44cd75fadba12; *Zhang, 2021*) and analyzed using Scanpy (https://scanpy.readthedocs.io/en/stable/). Muscle cells were extracted according to the expression of *Myod1* or *Pax7*.

## Human muscle transcriptome data

LSD1 mRNA expression in skeletal muscles of 55- to 80-year-old men and women were analyzed using a publicly available RNA-seq data (*Tumasian et al., 2021*). Expression values in TPM were used for linear regression analysis by JMP software (version 10.0.2d1).

## Statistical analyses

Sample size was determined based on our pilot experiments and previous studies in the related research area. Data are presented as the mean ± SD. Equality of variance was determined by using F test. If the variances were equal, two-tailed Student's t-test was used, and if not, two-tailed Welch's t-test was used.

## Materials availability statement

Newly created materials can be provided following the material transfer agreement of Kumamoto University.

## Acknowledgements

This work was supported by the following funding sources: JSPS KAKENHI (Grant Numbers: 20H04108 and 21K19513 [SH], 21H02686 and 20KK0185 [MN]), Takeda Science Foundation (SH and MN), Nakatomi Foundation (SH), the Japan Agency for Medical Research and Development (21gk0210029h0101 [SH]), Center for Metabolic Regulation of Healthy Aging, Kumamoto Univ. (HA), and the program of the Inter-University Research Network for High Depth Omics, IMEG, Kumamoto University (KE and SH). We would like to thank Editage (https://www.editage.com) for English language editing.

## Additional information

### Funding

| Funder | Grant reference number | Author |
| --- | --- | --- |
| Japan Society for the Promotion of Science | 20H04108 | Shinjiro Hino |

| Funder | Grant reference number | Author |
|---|---|---|
| the Inter-University Research Network for High Depth Omics, IMEG, Kumamoto University | | Shinjiro Hino |
| Japan Society for the Promotion of Science | 21K19513 | Shinjiro Hino |
| Japan Society for the Promotion of Science | 21H02686 | Mitsuyoshi Nakao |
| Japan Society for the Promotion of Science | 20KK0185 | Mitsuyoshi Nakao |
| Japan Agency for Medical Research and Development | 21gk0210029h0101 | Shinjiro Hino |
| Takeda Science Foundation | | Shinjiro Hino Mitsuyoshi Nakao |
| Nakatomi Foundation | | Shinjiro Hino |
| Center for Metabolic Regulation of Healthy Aging, Kumamoto University | | Hirotaka Araki |

The funders had no role in study design, data collection and interpretation, or the decision to submit the work for publication.

### Author contributions

Hirotaka Araki, Conceptualization, Data curation, Formal analysis, Funding acquisition, Investigation, Visualization, Writing - original draft; Shinjiro Hino, Conceptualization, Data curation, Formal analysis, Funding acquisition, Investigation, Methodology, Writing - original draft, Project administration, Writing - review and editing; Kotaro Anan, Conceptualization, Investigation, Methodology; Kanji Kuribayashi, Formal analysis, Investigation; Kan Etoh, Formal analysis, Visualization, Methodology; Daiki Seko, Methodology; Ryuta Takase, Kensaku Kohrogi, Investigation; Yuko Hino, Investigation, Methodology; Yusuke Ono, Supervision, Methodology; Eiichi Araki, Supervision; Mitsuyoshi Nakao, Conceptualization, Supervision, Funding acquisition, Investigation, Writing - original draft, Writing - review and editing

### Author ORCIDs
Hirotaka Araki http://orcid.org/0000-0001-8679-948X
Shinjiro Hino http://orcid.org/0000-0002-5748-016X
Kanji Kuribayashi http://orcid.org/0000-0003-2666-1141
Ryuta Takase http://orcid.org/0000-0003-2860-2647
Eiichi Araki http://orcid.org/0000-0002-4064-7525
Mitsuyoshi Nakao http://orcid.org/0000-0002-2196-8673

### Ethics

Animal experiments were approved by the Animal Care and Use Committee of Kumamoto University (Permit No. A2019-098), and conducted in accordance with its guidelines.

### Decision letter and Author response
Decision letter https://doi.org/10.7554/eLife.84618.sa1
Author response https://doi.org/10.7554/eLife.84618.sa2

## Additional files

### Supplementary files
• Supplementary file 1. Top 50 differentially regulated genes in dexamethasone (Dex)-treated lysine-specific demethylase-1 (LSD1)-mKO mice.
• Supplementary file 2. Cluster 6 genes in *Figure 5A*.

- Supplementary file 3. Primers used in this study.
- Supplementary file 4. Chromatin immunoprecipitation (ChIP) primers used in this study.
- Supplementary file 5. Full uncropped blots.
- MDAR checklist

## Data availability

RNA-seq data generated in this study were deposited in the Gene Expression Omnibus (accession No. GSE198911).

The following dataset was generated:

| Author(s) | Year | Dataset title | Dataset URL | Database and Identifier |
|-----------|------|---------------|-------------|-------------------------|
| Araki H, Hino S, Nakao M | 2022 | polyA RNA-seq of WT and LSD1-mKO mice under two different environmental stresses | https://www.ncbi.nlm.nih.gov/geo/query/acc.cgi?acc=GSE198911 | NCBI Gene Expression Omnibus, GSE198911 |

The following previously published dataset was used:

| Author(s) | Year | Dataset title | Dataset URL | Database and Identifier |
|-----------|------|---------------|-------------|-------------------------|
| Tumasian et al | 2014 | Skeletal muscle transcriptome in healthy aging | https://www.ncbi.nlm.nih.gov/geo/query/acc.cgi?acc=GSE164471 | NCBI Gene Expression Omnibus, GSE164471 |

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

# Appendix 1

## Appendix 1—key resources table

| Reagent type (species) or resource | Designation | Source or reference | Identifiers | Additional information |
|---|---|---|---|---|
| Strain, strain background (*Mus musculus*) | *Lsd1*-floxed mice (B6.129-*Kdm1a*^tm1.1Sho^/J) | Jackson Laboratories | Strain #:023969 RRID:IMSR_JAX:023969 | |
| Strain, strain background (*Mus musculus*) | Tg(*ACTA1-cre/Esr1\**)2Kesr/J | Jackson Laboratories | Strain #:025750 RRID:IMSR_JAX:025750 | |
| Strain, strain background (*Mus musculus*) | *Lsd1*^flox/flox^; Tg^ACTA1-CreERT2/+^ conditional knockout mice (LSD1-mKO mice) | This paper | N/A | See 'Animal studies' in Methods |
| Strain, strain background (*Mus musculus*) | C57BL6/J mice | Charles River Laboratories | N/A | |
| Cell line (*Mus musculus*) | C2C12 | RIKEN | RCB0987 | |
| Cell line (*Ruttus norvegicus*) | H9c2 | ATCC | CRL-1446 | |
| Antibody | Rabbit polyclonal anti-LSD1 | abcam | ab17721 | WB (1:500), ChIP (5 µg), Co-IP (3 µg) |
| Antibody | Rabbit polyclonal anti-Foxk1 | abcam | ab18196 | WB (1:1000), ChIP (5 µg), Co-IP (3 µg) |
| Antibody | Rabbit monoclonal anti-ERR gamma | abcam | ab128930 | WB (1:1000) |
| Antibody | Mouse monoclonal anti-MHC type I | DHSB | BA-F8 | IHC (1:50) |
| Antibody | Mouse monoclonal anti-MHC type IIA | DHSB | SC-71 | IHC (1:50) |
| Antibody | Rabbit polyclonal anti-Laminin | Sigma-Aldrich | L9393 | IHC (1:500) |
| Antibody | Rabbit polyclonal anti-Akt | Cell Signaling Technology | #9272 | WB (1:1000) |
| Antibody | Rabbit polyclonal anti-Phospho-Akt | Cell Signaling Technology | #9271 | WB (1:1000) |
| Antibody | Rabbit polyclonal anti-LC3 | MBL | PM036 | WB (1:1000) |
| Antibody | Mouse monoclonal anti-Phospho-4E-BP1 (Thr37/46) | Cell Signaling Technology | #2855 | WB (1:1000) |
| Antibody | Rabbit polyclonal anti-4E-BP1 | Cell Signaling Technology | 9454 | WB (1:1000) |
| Antibody | Mouse monoclonal anti-SQSTM1(p62) | Santa Cruz | sc-28359 | WB (1:1000) |
| Antibody | Rabbit polyclonal anti-GAPDH | Santa Cruz | sc-25778 | WB (1:6000) |
| Antibody | Rabbit polyclonal anti-trimethyl-Histone H3 (Lys4) | Millipore | 07-473 | ChIP (3 µg) |
| Antibody | Rabbit polyclonal anti-Histone H3 | abcam | ab1791 | WB (1:5000) |
| Recombinant DNA reagent | Control Double Nickase Plasmid | Santa Cruz Biotechnology | sc-437281 | |
| Recombinant DNA reagent | FOXK1 Double Nickase Plasmid | Santa Cruz Biotechnology | sc-437282-NIC | |
| Recombinant DNA reagent | pENTR4-H1 | RIKEN BRC DNA BANK | RDB04395 | |
| Recombinant DNA reagent | pCAG-HIVgp | RIKEN BRC DNA BANK | RDB04394 | |
| Recombinant DNA reagent | pCMV-VSV-G-RSV-Rev | RIKEN BRC DNA BANK | RDB04393 | |
| Recombinant DNA reagent | CS-RfA-EVBsd | RIKEN BRC DNA BANK | RDB06090 | |

*Appendix 1 Continued on next page*

*Appendix 1 Continued*

| Reagent type (species) or resource | Designation | Source or reference | Identifiers | Additional information |
|---|---|---|---|---|
| Recombinant DNA reagent | CS-RfA-EVBsd shFoxk1 | This paper | N/A | See 'Construction of lentiviral expression vectors and their transduction' in Methods |
| Recombinant DNA reagent | CS-RfA-EVBsd shGL3 | This paper | N/A | See 'Construction of lentiviral expression vectors and their transduction' in Methods |
| Recombinant DNA reagent | CSII-EF-3xFLAG-hLSD1-Bsd | This paper | N/A | A lentiviral construct used for the forced expression of human LSD1 |
| Sequence-based reagent | Primers for qRT-PCR | This paper | N/A | See *Supplementary file 3* |
| Sequence-based reagent | See Table S2 for primers for ChIP-qPCR | This paper | N/A | See *Supplementary file 4* |
| Commercial assay or kit | NEBNext Ultra DNA Library Prep Kit | New England Biolabs | E7370S | |
| Commercial assay or kit | NEBNext Poly(A) mRNA Magnetic Isolation Module | New England Biolabs | E7490S | |
| Chemical compound, drug | Tamoxifen | Sigma-Aldrich | T5648 | |
| Chemical compound, drug | Dexamethasone | Sigma-Aldrich | D4902 | |
| Chemical compound, drug | Insulin | nacalai tesque | 19251-24 | |
| Chemical compound, drug | Streptozotocin | Wako | 197-15153 | |
| Chemical compound, drug | T-3775440 hydrochloride | MedChemExpress | HY-103085 | |
| Chemical compound, drug | FuGENE 6 Transfection Reagent | Promega | E2691 | |
| Chemical compound, drug | Puromycin dihydrochloride | Sigma-Aldrich | P8833 | |
| Chemical compound, drug | Blasticidin S | FUJIFILM Wako Chemicals | 029-18701 | |
| Chemical compound, drug | Can Get Signal immunostain Solution A | TOYOBO | NKB-501 | |
| Chemical compound, drug | malinol | Muto Pure Chemicals | 2009-1 | |
| Chemical compound, drug | Can Get Signal Solution 1 | TOYOBO | NKB-201 | |
| Chemical compound, drug | Can Get Signal Solution 2 | TOYOBO | NKB-301 | |
| Chemical compound, drug | TRIzol RNA Isolation Reagent | Thermo Fisher Scientific | 15596026 | |
| Chemical compound, drug | ReverTra Ace qPCR RT Master Mix | TOYOBO | FSQ-201 | |
| Chemical compound, drug | THUNDERBIRD SYBR qPCR Mix | TOYOBO | QPS-201 | |
| Software, algorithm | BZ-X Analyzer software | KEYENCE | N/A | |
| Software, algorithm | Homer v4.9.1 | *Heinz et al., 2010* | http://homer.ucsd.edu/homer/ | |
| Software, algorithm | STAR | *Dobin et al., 2013* | https://github.com/alexdobin/STAR; *Dobin, 2023* | |
| Software, algorithm | Edge R | *Robinson et al., 2010* | http://bioconductor.org/packages/release/bioc/html/edgeR.html | |
| Software, algorithm | Heatmapper | *Babicki et al., 2016* | http://www.heatmapper.ca/expression/ | |

*Appendix 1 Continued on next page*

*Appendix 1 Continued*

| Reagent type (species) or resource | Designation | Source or reference | Identifiers | Additional information |
|---|---|---|---|---|
| Software, algorithm | DEseq2 | *Love et al., 2014* | https://bioconductor.org/packages/release/bioc/html/DESeq2.html | |
| Software, algorithm | ComplexHeatmap | *Gu et al., 2016* | https://github.com/jokergoo/ComplexHeatmap; *Gu, 2022* | |
| Software, algorithm | Molecular Signature Database v7.4 | *Subramanian et al., 2005* | https://www.gsea-msigdb.org/gsea/msigdb/ | |
| Software, algorithm | HOMER V4.9.1 | *Heinz et al., 2010* | http://homer.ucsd.edu/homer/ | |
| Software, algorithm | JMP (version 10.0.2d1) | SAS Institute Inc | N/A | |
| Software, algorithm | GraphPad Prism version 8.4.3 | GraphPad Software | N/A | |
| Software, algorithm | Integrative Genomics Viewer (IGV) | James T et al., 2011 | https://software.broadinstitute.org/software/igv/ | |

