## [Editor Report]

This paper investigates the role of lysine-specific histone demethylase 1 (LSD1) in skeletal muscle responses to stresses. The findings link LSD1 to responses of skeletal muscle to glucorticoids and exercise in a fiber-type-dependent manner. The paper should be of broad interest to those studying muscle biology and physiology.

---

## [Decision Letter]

[Editors' note: this paper was reviewed by Review Commons.]

---

## [Author Response]

General Statements [optional]

We thank the reviewers for productive comments. Many of the experiments and analyses suggested have already been carried out. For the remaining experiments, we have already gathered necessary materials. Thus, we can add highly relevant revisions to the next version within a short period.

Description of the planned revisionsInsert here a point-by-point reply that explains what revisions, additional experimentations and analyses are planned to address the points raised by the referees.Reviewer #1:Major #1This study primarily uses the genetic mouse model in which LSD1 gene is inactivated after tamoxifen injection in 8 weeks old mice, as shown in supplemental figure 1 B and C. 8 weeks after birth postnatal growth of muscle is not complete and the contribution of satellite cells to muscle growth is still significant. Therefore the timing of tamoxifen injection used cannot discriminate if the observed phenotype involves the function of LSD1 during the post-natal growth of the muscle or in the muscle fibers or both. One way to demonstrate the real contribution of LSD1 in the maintenance of muscle fibers plasticity under environmental stress would be to inject Tamoxifen later (around 10-12 weeks of age), in order to remove a possible bias caused by the contribution of satellite cells during the post-natal growth. At least key findings should be confirmed at this later stage.

In this study, we used ACTA1-CreERT mice to conditionally knockout LSD1 in the skeletal muscle. The ACTA1 promoter is derived from human muscle actin gene, which is not expressed in the satellite cells, and has been widely used for the transgene expression in myofibers (Stantzou *et al.* Development 2017). Thus, the inactivation of LSD1 occurs in the existing myofibers, and alterations in satellite cell function, if any, would be indirect effects of the loss of LSD1 in mature myocytes or differentiating myoblasts.

To test whether postnatal muscle growth was affected in our LSD1-mKO mice, we administrated tamoxifen (4OHT) to pre-weaning mice (11 days old). LSD1 depletion did not affect the expression of muscle fiber genes, when muscle tissues were isolated from mice 11 days after the start of 4OHT (Figure 1—figure supplement 2).

These evidences exclude the contribution of satellite cells in the phenotypes observed in the LSD1-mKO mice. Figure 1—figure supplement 2 has been added in the revised manuscript, and described in the Results section (P.5).

Major #2LSD1 m-KO muscles seem to have more type I and IIA fibers than WT, even without DEX treatment. Is it possible to quantify the results in supplemental figure 4C?

As suggested, we quantitatively analyzed the fiber type compositions in Figure 1—figure supplement 6C (former Figure S4C) using the data from WT (n=4) and LSD1-mKO (n=5) mice (Figure 1—figure supplement 6D ). We did not find a significant difference between these mice, confirming our finding that the loss of LSD1 accelerates the Dex-driven phenotypic changes. We also analyzed the Soleus muscle from untreated mice, and found no obvious difference between the WT and KO (Figure 1—figure supplement 7A-B ).

Major #3The effect on fiber type is convincing, while variations in gene expression are of quite low amplitude. However, the atrophy should be induced by other means to ensure that the effects are specific to GC/nuclear receptors pathways; Denervation? Starvation? Not all the experiments need to be repeated, just key results such as, for example, exacerbation of atrophy in LSD1 m-KO, Foxk1 increase.

We agree that testing alternative atrophy models is important for generalizing our findings. For this, we employed a model for diabetes-related muscle atrophy. A pro-diabetic agent streptozotocin (STZ) disturbs the function of pancreatic islet leading to fast-fiber atrophy (O’Neill *et al.* Diabetes 2019). LSD1-mKO did not affect the muscle weight in STZ-treated mice (Figure 2—figure supplement 3). Consistently, there were no major difference in the expression of atrophy genes in STZ-treated WT and LSD1-mKO mice (Figure 2—figure supplement 3 ). These results suggest that the LSD1 function depends on the source of atrophy-inducing stress, and that the loss of LSD1 sensitized the muscle to GC-mediate signaling.

Major #4Autophagy data: the effect on the LC3I/LC3II ratio are modest. The autophagy part should be removed or completed with additional data to convincingly show that autophagy is affected. Links between LSD1 and mTOR have been published, so the mTOR pathway could be investigated in the model (S6k, S6 and 4EBP1 phosphorylation). Given AKT levels and phosphorylation are affected by the absence of LSD1 + DEX, it can be predicted that mTOR activity will change.

We have analyzed the expression of additional autophagy markers p62 and phosphorylated 4EBP1. Consistent with the upregulated expression of atrophy genes and increased LC3I/II ratio, LSD1-mKO mice had elevated levels of p62 and phosphorylated 4EBP1 (Figure 1—figure supplement 5B and D,). Altogether, the data suggest that Dex-induced muscle atrophy was exacerbated by the loss of LSD1. These data are added as Figure 1—figure supplement 5B and D, and described in the Results section (P.6).

Major #5The ability of LSD1 to retain FOXK1 in the nucleus is an important information that should be better supported experimentally. In the absence of such information, no mechanism can be proposed for the effect of LSD1 of FOXK1. The immunofluorescence images provided are not convincing and moreover they could be interpreted by a reduction in the level of FOXK1 protein (degradation?) rather than by a nuclear exclusion in the presence of DEX. This point should be addressed, authors could include western blot of nuclear and cytoplasmic fractions to better quantify the nuclear level of FOXK1 in absence of LSD1.

As suggested, we performed WB analyses of Foxk1 using nuclear and cytoplasmic fractions. Our results show that LSD1 inhibition combined with Dex treatment led to a reduced nuclear/cytoplasmic ratio (Figure 3—figure supplement 1B ). This data is consistent with our immunofluorescence data, and altogether suggest that LSD1 promotes the nuclear retention of Foxk1. The data has been added as Figure 3—figure supplement 1B and described in the Results section (P.9).

Major #6The absence of centralized nuclei indicates that there is no fiber regeneration but it does not exclude the possibility that satellite cells were recruited to existing fibers and thus participated to hypertrophy. To eliminate this possibility, the average nuclei/cytoplasm volume should decrease if hypertrophy results from increased protein synthesis and not myonuclei accretion. This should be checked.

We histologically analyzed the sections of Gas muscles after Dex treatment and found that there is no evidence of central nuclei in either WT or KO mice (Figure 1—figure supplement 7C).

As mentioned above (Major #1), it is unlikely that the satellite cell function was responsible for the enhanced atrophic phenotype.

Major #7The upregulation of ERRγ in the absence of LSD1 is convincing in the VWR conditions. ERRγ level should be evaluated in the sedentary LSD1 KO mice.

We have analyzed the expression of ERRg in sedentary mice, and found no significant difference between WT and KO mice (Figure 5—figure supplement 1A ). This suggests that the loss of LSD1 in combination with VWR training led to the increased expression of ERRg. This data is included in the revised manuscript as Figure 5—figure supplement 1A and described in the Results section (P.12).

Minor #1There is a clear difference in the number of mouse replicates between treated (Dex or VWR) and non-treated mice, regardless the genotype. Experiments with non-treated mice lack adequate numbers to make a definitive conclusion. For example, there is a huge spread in the data in Figure 1 B and 4 D. If the number of animals would have been increased, would the absence of difference hold up?

We increased the number of non-treated animals in Figures 1B and 4B as suggested. Nonetheless, we did not find any significant differences in the muscle weight (Figures 1B and 4B). These changes are reflected on original Figures 1B and 4B. For Figure 1—figure supplement 4A-B, we used violin plots to clearly visualize the differences among groups.

Minor #2The authors claim that: "Consistent with the results of the augmented endurance capacity, the Sol muscle in the KO mice showed enhanced succinate dehydrogenase (SDH) staining, indicating that the number of oxidative fibers increased (Figure 4F and Supplemental Figure 8F)". However, supplemental figure 8 D indicates that the number of type I fibers does not change compared to WT. Authors should clarify this statement.

Indeed, we found that the area of type I fiber but not the number was increased in the LSD1-KO Sol (Figure 4D and Figure 4—figure supplement 1D). Because SDH staining reflects the OXPHOS capacity in all fiber types, it is possible that the OXPHOS capacity in the fibers other than type I had been augmented by LSD1-KO. Thus, for clarification, we will change the statement as follows: OXPHOS capacity of Sol was enhanced by the loss of LSD1 (Results section, P.11)**.**

Reviewer #2Methods1. The authors used the Cre-lox system with tamoxifen to generate skeletal muscle-specific LSD1 KO mice. It is clear that both the mRNA and protein levels of LSD1 in various muscles were dramatically reduced, but there is still some LSD1 expressed in skeletal muscle, especially in Sol muscle (Supplemental Figure 1C). The author needs to think about whether it is appropriate to use the term "LSD1 knockout" or "LSD1 deficiency".

We thank the reviewer for this comment. In this study, we crossed LSD1-floxed mice with ACTA1-creERT mice. This enables the deletion of critical exons of *LSD1* in mature myocytes and myogenic precursors that have initiated the differentiation program. LSD1 is a ubiquitously expressed gene, and it is known that immature myogenic cells (*e.g.,* satellite cells, Tosic et al. Nat Commun. 2018) and other non-myogenic cells such as hematopoietic and vascular cells abundantly express LSD1 (Kerenyi et al. *ELife* 2013, Yuan et al. Biochem Pharmacol. 2022). Thus, it is likely that LSD1 expression by these cell types were detected in our whole muscle western blots. We added these statements in the text for clarification (Results, P.5).

Results2. To identify the transcriptional regulators that mediate the regulation of atrophy-associated genes by LSD1, the authors performed motif analyses on the promotor regions of upregulated genes in LSD1-mKO Gas. Based on the results and other reports, they focused on Foxk1 and proved LSD1 and Foxk1 cooperatively regulate the atrophy transcriptome in the presence of Dex. However, Figure 3C showed that a transcription factor Nfatc1 is also reduced in Sol muscle similar to Foxk1. Also, other studies demonstrated that the transcription factor NFATc1 controls fiber type composition and is required for fast-to-slow fiber type switching in response to exercise in vivo. More specifically, NFATc1 inhibits MyoD-dependent fast fiber gene promoters by physically interacting with the N-terminal activation domain of MyoD and blocking recruitment of the essential transcriptional coactivator p300 (Cell Rep. 2014 Sep 25; 8(6): 1639-1648). Furthermore, it has been reported that LSD1 Controls Timely MyoD Expression via MyoD Core Enhancer Transcription (Cell Rep. 2017 Feb 21;18(8):1996-2006. doi: 10.1016/j.celrep.2017.01.078). It is unclear how the authors exclude Nfatc1 for the LSD1-mediated effects in different muscle fibers. Further experiments may be necessary to exclude Nfatc1.

We thank the reviewer for an insightful comment. In addition to Foxk1, we tested the involvement of NFATc1 in the gene regulation under LSD1-depleted state. We treated C2C12 with an LSD1 inhibitor S2101 in combination with a calcium ionophore that promotes the transcriptional function of NFATc1 by inducing its nuclear localization (Meissner et al. J Cell Physiol. 2007). While LSD1 inhibition promoted the expression of Pgc1a and Myh7, ionophore treatment had no additive effects. Because we found a physical association of Foxk1 with LSD1, we focused on the functional involvement of Foxk1 in LSD1-mediated repression of atrophy genes. We recently performed an ATAC-seq analysis in Dex-treated muscle, and found that the Foxk1 motif but not the NFATc1 motif was enriched in the LSD1-KO-specific open chromatin regions. This data further suggests the significant contribution of Foxk1 in the transcriptional regulation under LSD1 depletion.

3. In figure 3D, only merged images were colored. It would be better to show colored images for Foxk1 and DAPI.

We replaced the images with the colored ones in Figure 3D.

4. Immunofluorescence analysis in C2C12 myotubes showed that Dex exposure reduced the nuclear retention of Foxk1, which was further promoted by the addition of T-3775440, an LSD1 inhibitor (Figure 3D). The author also used Foxk1-KO C2C12 myotubes to prove LSD1 and Foxk1 cooperation to regulate the expression of type I /IIA fiber and atrophy genes in Foxk1-KO cells. Are the effects of LSD1 dependent on Foxk1 or synergistically acting with Foxk1? The treatment of LSD1 inhibitor in Foxk1-KO C2C12 may be helpful to answer this question.

As suggested, we tested the combination effect of LSD1 inhibition and Foxk1-KO. The treatment of Foxk1-KO cells with T-3775440 did not show additional effects on the expression of atrophy and slow fiber genes (Figure 3—figure supplement 2B). Next, we analyzed the chromatin association of LSD1 in Foxk1-KO cells by a ChIP assay, and found that the occupancy of LSD1 at atrophy gene loci was diminished in Foxk1-KO cells (Figure 3G and Figure 3—figure supplement 2C). These results collectively suggest that the LSD1 largely depend on Foxk1 to control atrophy genes. These data are included in the revised version as Figure 3G and Figure 3—figure supplement 2B-C, and described in the text (Results, P. 10).

5. In supplementary figure 2, body weight in the mKO+Dex group was reduced in comparison to that of WT+Dex. How about the body weight of mKO mice without Dex injection compared to that of WT? This data will be helpful to understand the effect of muscle-specific LSD1 deficiency on whole-body energy balance.

We measured the body weight of untreated mice, and found that there is no genotype effect (Figure 1—figure supplement 3D-E ). Thus, we think that LSD1-mKO alone does not influence the whole-body energy balance. We included this data as Figure 1—figure supplement 3D-E.

6. The authors analyzed the size distribution of myofibers and mentioned that large type I and type IIA fibers preferentially increased in the LSD1-mKO muscle, whereas large type IIB + IIX fibers decreased (Supplemental Figure 4, B, E, and F). It is better to show the results of statistics. If no significance were found, it should be mentioned in the result section.

We have performed statistical analyses on Figure 1—figure supplement 7D-E (Supplemental Figure 4E and F in the previous version), and found that a fraction of large type I fibers was significantly larger in KO mice. This result is indicated in the figure.

7. Page 11, To reveal the genes regulated by LSD1 under the VWR condition, the authors performed additional RNA-seq analysis using Sol muscle. The non-hierarchical clustering analysis was informative and showed signaling pathways related to ‘mitochondrion’, ‘mitochondrion organization’, and ‘oxidative phosphorylation’ were altered in the Sol muscle deficient in LSD1 under the VWR condition (Figure 5B). However, it is unclear why they focus on Err-γ to explain LSD1-KO phenotypes in Sol muscle. Is this gene also derived from RNA seq? It is better to show whether Err-γ expression is also significantly altered based on RNA seq data.

Indeed, ERRg was upregulated by LSD1-KO+VWR and was included in the Cluster 6 genes together with the OXPHOS and mitochondria-related genes (Figure 5A). These data prompted us to focus on ERRg as a potential factor that explains the LSD1-KO phenotype.

8. The authors claim that LSD1 serves as an "epigenetic barrier" that optimizes fiber type-specific responses and muscle mass under stress conditions. This claim is derived from the loss of function studies. To generalize the functions of LSD1, the gain of function studies will be also necessary. Adding the characteristics of LSD1 overexpression in C2C12 cells will further improve the quality of the manuscript.

We agree that the gain of function further strengthens the quality of our study. As suggested, we introduced LSD1 into C2C12 cells using a lentivirus vector. However, the net expression of endo- and exogenous LSD1 (mouse and human, respectively) were similar between mock- and LSD1-introduced cells (data not shown). This suggests that LSD1 protein level was maintained at a certain level in these cells.

Because *Esrrg* is completely silenced in C2C12 cells, it is difficult to monitor ERRg-mediated gene regulation in these cells. To overcome this, we made use of a rat cardiomyocyte cell line H9c2, in which ERRg is functionally involved in differentiation (Sakamoto *et al.* Nat Commun 2022). We first inhibited LSD1 using the inhibitor T-3775440 in these cells, and found that the expression of *Esrrg* and its downstream target *Perm1* was significantly upregulated (Figure 5C, left). We next overexpressed LSD1 in these cells and found a downregulation of *Esrrg* and *Perm1* (Figure 5C, right). These data are highly consistent with our in vivo data that LSD1 represses *Esrrg* in slow fiber enriched muscle. These data are added as Figure 5—figure supplement 1C-E, and described in the Results section (P. 12-13).

Discussion9. The authors mentioned supplementary figure 10 only at the end of the manuscript of the Discussion section (page 15) without a specific explanation of the figures in the result section. The data are important in that LSD1 expression in human muscles declined with age and showed a negative correlation with the expression of the atrophy gene. It should be presented in the result section with a more detailed description.

We agree that these data are important and need further explanations. We added the description in details in the Results section (P.13) and moved the entire figure to the main figure (Figure 6).

10. There are other studies to examine LSD1 and muscle regeneration or functions (e.g. Nat Commun 9, 366 (2018). https://doi.org/10.1038/s41467-017-02740-5). More discussion to compare the current study and other studies will be necessary.

We thank the reviewer for this comment. We added a discussion about the phenotypic difference between our mice and the previously reported LSD1-KO mice (Tosic et al. 2018) (Discussion, P.14).

Description of the revisions that have already been incorporated in the transferred manuscriptPlease insert a point-by-point reply describing the revisions that were already carried out and included in the transferred manuscript. If no revisions have been carried out yet, please leave this section empty.

As described above, a number of experiments and analyses requested by the reviewers have already been done. Those will be incorporated in the next version.

Description of analyses that authors prefer not to carry outPlease include a point-by-point response explaining why some of the requested data or additional analyses might not be necessary or cannot be provided within the scope of a revision. This can be due to time or resource limitations or in case of disagreement about the necessity of such additional data given the scope of the study. Please leave empty if not applicable.

As described above, for some comments, there are alternative solutions other than the ones requested by the reviewers.

Rev#1, Major#1, Major#6

In addition to the experiments using young (8wks) mice, we performed experiments using pre-weaning mice (11days), in which satellite cells actively contribute to the muscle growth. In this experiment, there were no differences in the muscle weight between WT and KO mice. In addition, our ACTA1-creERT-mediated LSD1 depletion system, the LSD1 gene was inactivated only after the onset of differentiation. From these evidences, we concluded that satellite cells are not the primary source of the LSD1-KO phenotype.

Thus, we do not think it necessary to perform experiments with older mice or reanalyze the nuclei/cytoplasm volume to test the possible contribution of satellite cells to the phenotype.